# Structure-guided glyco-engineering of ACE2 for improved potency as soluble SARS-CoV-2 decoy receptor

Tümay Capraz[1†], Nikolaus F Kienzl[2†], Elisabeth Laurent[3†], Jan W Perthold[1], Esther Föderl-Höbenreich[4], Clemens Grünwald-Gruber[5], Daniel Maresch[5], Vanessa Monteil[6], Janine Niederhöfer[7], Gerald Wirnsberger[7], Ali Mirazimi[6,8], Kurt Zatloukal[4], Lukas Mach[2]*, Josef M Penninger[9,10]*, Chris Oostenbrink[1]*, Johannes Stadlmann[5,9]*

[1]Institute for Molecular Modeling and Simulation, University of Natural Resources and Life Sciences (BOKU), Vienna, Austria; [2]Institute of Plant Biotechnology and Cell Biology, Department of Applied Genetics and Cell Biology, University of Natural Resources and Life Sciences (BOKU), Vienna, Austria; [3]Institute of Molecular Biotechnology, Department of Biotechnology and Core Facility Biomolecular & Cellular Analysis, University of Natural Resources and Life Sciences (BOKU), Vienna, Austria; [4]Diagnostic and Research Institute of Pathology, Medical University of Graz, Graz, Austria; [5]Institute of Biochemistry, Department of Chemistry, University of Natural Resources and Life Sciences, Vienna, Austria; [6]Karolinska Institute, Department of Laboratory Medicine, Stockholm, Sweden; [7]Apeiron Biologics, Vienna, Austria; [8]National Veterinary Institute, Uppsala, Sweden; [9]IMBA - Institute of Molecular Biotechnology of the Austrian Academy of Sciences, Dr. Bohr, Vienna, Austria; [10]Department of Medical Genetics, Life Sciences Institute, University of British Columbia, Vancouver, Canada

*For correspondence:
lukas.mach@boku.ac.at (LM);
josef.penninger@ubc.ca (JMP);
chris.oostenbrink@boku.ac.at
(CO);
j.stadlmann@boku.ac.at (JS)

†These authors contributed equally to this work

**Abstract** Infection and viral entry of SARS-CoV-2 crucially depends on the binding of its Spike protein to angiotensin converting enzyme 2 (ACE2) presented on host cells. Glycosylation of both proteins is critical for this interaction. Recombinant soluble human ACE2 can neutralize SARS-CoV-2 and is currently undergoing clinical tests for the treatment of COVID-19. We used 3D structural models and molecular dynamics simulations to define the ACE2 N-glycans that critically influence Spike-ACE2 complex formation. Engineering of ACE2 N-glycosylation by site-directed mutagenesis or glycosidase treatment resulted in enhanced binding affinities and improved virus neutralization without notable deleterious effects on the structural stability and catalytic activity of the protein. Importantly, simultaneous removal of all accessible N-glycans from recombinant soluble human ACE2 yields a superior SARS-CoV-2 decoy receptor with promise as effective treatment for COVID-19 patients.

## Editor's evaluation

This paper takes on the important topic of therapeutics for SARS-CoV-2. This is an early-phase proof of concept study showing that a de-glycosylated form of the virus receptor, ACE2, could serve to reduce virus replication by acting as a decoy receptor that doesn't lead to infection.

## Introduction

The rapid spread of Severe Acute Respiratory Syndrome Coronavirus-2 (SARS-CoV-2), the causative pathogen of human coronavirus disease 2019 (COVID-19), has resulted in an unprecedented pandemic and worldwide health crisis. Similar to the beta-coronaviruses SARS-CoV and Middle Eastern Respiratory Syndrome (MERS)-CoV, SARS-CoV-2 is highly transmissible and can lead to lethal pneumonia and multi-organ failure (*Zhu et al., 2020*). For infection and viral entry, the Spike surface protein of SARS-CoV-2 binds to angiotensin converting enzyme 2 (ACE2) on host cells (*Wu et al., 2020*; *Zhou et al., 2020*). Recombinant soluble human ACE2 (rshACE2) has been shown to bind Spike (*Wang et al., 2020*), can effectively neutralize SARS-CoV-2 infections (*Monteil et al., 2020*, *Monteil et al., 2021*), and the corresponding drug candidate APN01 has undergone a phase II clinical trial for the treatment of hospitalized cases of COVID-19 (ClinicalTrials.gov Identifier: NCT04335136). A first case study of its use in a patient has been reported recently (*Zoufaly et al., 2020*). Additionally, an aerosol formulation of APN01 has been developed and is currently undergoing Phase I clinical studies.

Multiple other therapeutic strategies attempt to target the Spike-ACE2 interaction, for example by development of neutralizing antibodies blocking the ACE2-binding site (*Barnes et al., 2020*) or lectins that bind to glycans on the Spike surface (*Chan et al., 2021*, *Hoffmann et al., 2021*). Using soluble ACE2 as a decoy receptor for Spike is particularly attractive, as it minimizes the risk that variants of concern may evade the treatment through mutations as has been observed for antibodies (*Weisblum et al., 2020*; *Li et al., 2021*; *Yuan et al., 2021*). Furthermore, protein engineering has yielded ACE2 variants with substantially improved affinities for Spike (*Glasgow et al., 2020*; *Chan et al., 2020*). Hence, soluble ACE2-based therapeutics offer considerable advantages over other therapeutic formats that aim to hamper the Spike-ACE2 interaction sterically.

Modern structural biology has been amazingly fast to respond to this pandemic. A mere 3 months after identification of SARS-CoV-2 as the etiologic agent of COVID-19, structures of the complex between ACE2 and the receptor binding domain (RBD) (*Wang et al., 2020*; *Lan et al., 2020*; *Yan et al., 2020*) and of the ectodomain of trimeric Spike (*Walls et al., 2020*; *Wrapp et al., 2020*) were already solved by X-ray crystallography or cryo-electron microscopy. While this provided unprecedented insight into the protein-protein interactions between Spike and ACE2, the structural impact of protein-bound glycans on the Spike-ACE2 interface could not be assessed experimentally so far due to their compositional diversity and conformational flexibility. Here, in silico modeling of the glycans offers a powerful alternative to study the effects of individual Spike and ACE2 glycans on the molecular interactions between these two proteins.

The SARS-CoV-2 Spike protein is heavily glycosylated with both complex and oligo-mannosidic type N-glycans (*Zhu et al., 2020*, *Hoffmann et al., 2021*; *Watanabe et al., 2020*; *Zhao et al., 2020*), thereby shielding a large portion of the protein surface (*Casalino et al., 2020*; *Sikora et al., 2021*; *Zimmerman et al., 2021*). Similarly, ACE2 is a glycoprotein with up to seven highly utilized sites of N-glycosylation (*Zhao et al., 2020*). Recent computational studies started to investigate protein glycosylation in the context of the interaction between Spike and ACE2 (*Zhao et al., 2020*; *Casalino et al., 2020*; *Mehdipour and Hummer, 2021*). Extensive all-atom molecular dynamics (MD) simulations indicated that Spike N-glycans attached to N165 and N234 could be important stabilizers of the ligand-accessible conformation of the receptor binding domain (RBD) (*Casalino et al., 2020*). Furthermore, it has been proposed that the N-glycan at position N343 acts as a gate facilitating RBD opening (*Sztain et al., 2021*). Other MD studies concluded that the glycans attached to N90 and N322 of ACE2 could be major determinants of Spike binding (*Mehdipour and Hummer, 2021*), while yet other simulation works postulate that glycosylation does not affect the RBD-ACE2 interaction significantly (*Cong et al., 2021*; *Delgado et al., 2021*). Genetic or pharmacological blockade of N-glycan biosynthesis at the oligomannose stage in ACE2-expressing target cells was found to dramatically reduce viral entry (*Yang et al., 2020*), even though several glycoforms of ACE2 were found to display comparatively moderate variation with respect to Spike binding (*Allen et al., 2021*). Hence, a detailed understanding on how individual glycans on both Spike and ACE2 influence their interaction and a comprehensive experimental validation of the MD findings is crucial for the rational design of novel therapeutic soluble ACE2 variants with enhanced Spike binding affinity and the capacity to block viral entry more efficiently than the native enzyme (*Zhao et al., 2020*). The identification of the Spike glycans essential for efficient association with ACE2 will be also critical to guide rational design of improved SARS-CoV-2 vaccines.

## Results

We started our research by creating 3D models of the trimeric Spike in complex with human ACE2 (hACE2). The RBD of Spike exists in two distinct conformations, referred to as 'up' and 'down' (*Walls et al., 2020*; *Wrapp et al., 2020*). The 'up' conformation corresponds to the receptor-accessible state with the RBD of one monomer exposed. By superimposing the RBD from the RBD-hACE2 complex (*Yan et al., 2020*) with the single RBD in the 'up' conformation (monomer 3) of the trimeric Spike (*Walls et al., 2020*), an initial model was obtained. Of note, although cryo-EM structures with more than one RBD in the 'up' conformation were shown to bind to two separate hACE2 molecules (rather than one single hACE2 dimer) (*Benton et al., 2020*), we decided to study the effect of the glycans on the Spike hACE2 interaction using the model with a single RBD in the 'up' conformation. To assess the impact of all seven individual N-glycosylation sites of hACE2 on its interaction with Spike, we first elucidated the entire glycome of rshACE2 (*Figure 1—figure supplement 1*). This also provided information on the glycans attached to N690, a glycosylation site not covered in previous glycoproteomic studies of soluble hACE2 (*Zhao et al., 2020*; *Allen et al., 2021*). For recombinant trimeric

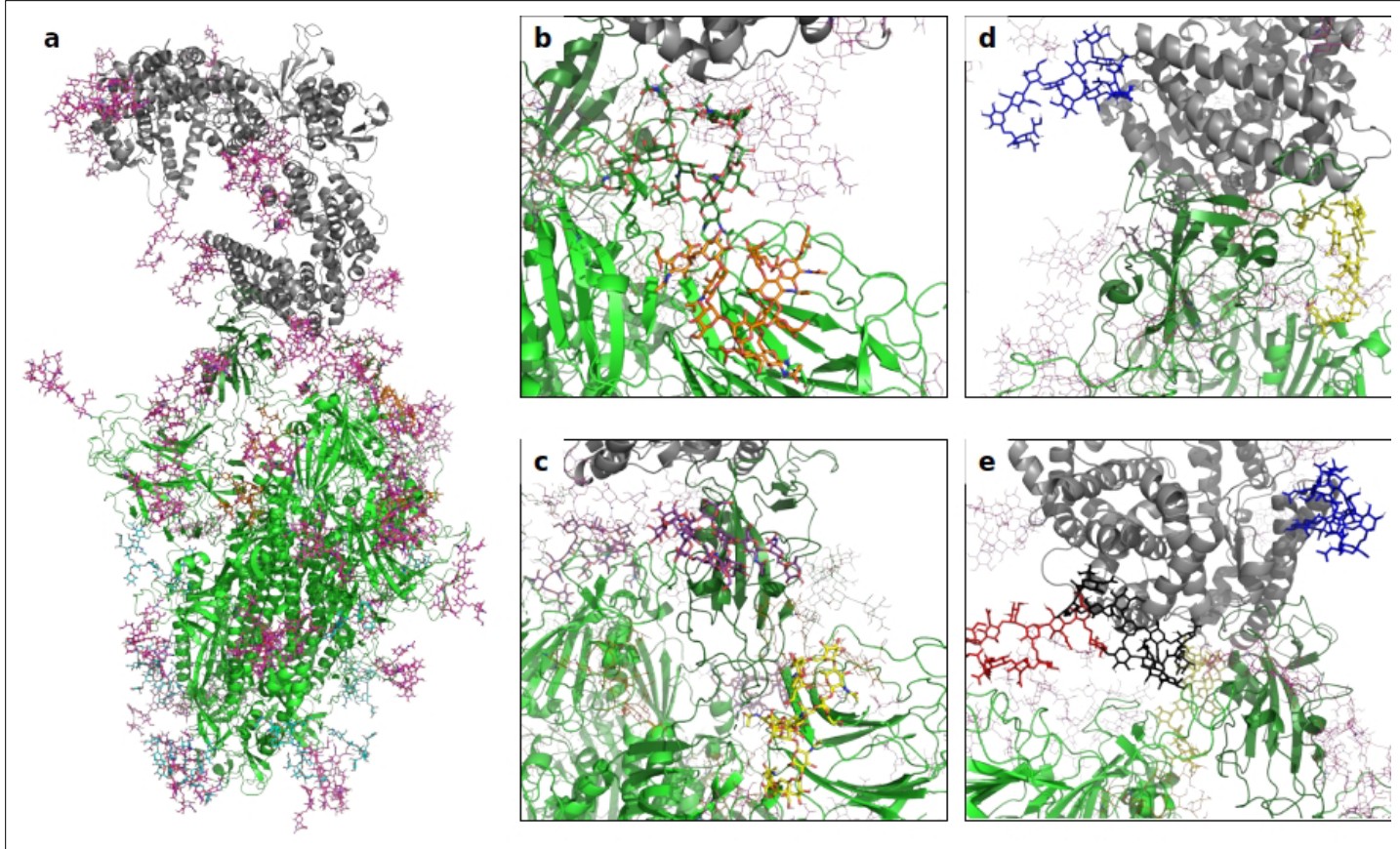

**Figure 1.** A 3D structural model of the glycosylated Spike-hACE2 complex. (**a**) 3D model of the Spike trimer (in green, with RBD of monomer three in dark green) binding to ACE2 (in gray) with complex glycosylation in magenta, Man5 glycans in light blue and Man9 glycans in orange. (**b**) Close-up view of the glycans at N122 (orange sticks) and N165 (dark green sticks) on monomer 3 of Spike. (**c**) Close-up view of the glycans at N331 (yellow sticks) and N343 (purple sticks) on monomer 3 of Spike. (**d**) Close-up view of the glycans at N53 (blue sticks) and N90 (yellow sticks) on ACE2. (**e**) Close-up view of the glycans at N53 (blue sticks), N90 (yellow sticks), N322 (black sticks), and N546 (red sticks).

The online version of this article includes the following figure supplement(s) for figure 1:

**Figure supplement 1.** Site-specific glycosylation profiles of rshACE2.

**Figure supplement 2.** Distance heatmap of human ACE2 (hACE2) residues in contact with Spike RBD.

**Figure supplement 3.** Electrostatic potential of the binding interface region on ACE2.

**Figure supplement 4.** Positional modeling of glycan N234 in the Spike trimer.

**Figure supplement 5.** Residue T324 of ACE2 at the interface to Spike.

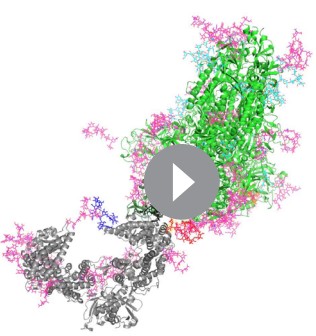

**Video 1.** Molecular dynamics simulation of trimeric Spike bound to ACE2. The movie highlights a three nano-second time segment of the molecular dynamics simulation (from 25 to 28 ns). Trimeric Spike is shown in green, RBD in dark green, and human ACE2 in gray. Complex glycosylation is shown in magenta, Man5 N-glycans in light blue and Man9 N-glycans in orange. Glycans of ACE2 at N53, N90, N322, and N546 are shown in blue, yellow, black, and red, respectively. https://elifesciences.org/articles/73641/figures#video1

Spike, the glyco-analysis has been reported elsewhere (*Hoffmann et al., 2021*; *Watanabe et al., 2020*; *Zhao et al., 2020*; *Sun et al., 2021*). Based on the site-specific glycosylation profiles, we added complex or oligo-mannosidic glycan trees to the respective sites of Spike and ACE2 (Supplementray File 1). We hence constructed fully glycosylated atomistic models of the trimeric Spike glycoprotein, free dimeric ACE2 and of the Spike glycoprotein in complex with dimeric hACE2 (*Figure 1*). As glycans are known to be particularly flexible, and these were modeled in a single low-energy conformation, we subsequently performed molecular dynamics simulations of the Spike-ACE2 complex (*Video 1*), and of free hACE2. This allowed us to study the conformational distribution of the glycans on the surface of the proteins and their dynamic effects on the interaction between Spike and hACE2. Inspection of the most important interacting residues on Spike and ACE2, their average distances and the electrostatic potential of the interface area identified critical contact sites (*Figure 1—figure supplements 2 and 3*).

We next quantified the complete solvent-accessible surface area (SASA) of the Spike protein in complex with ACE2, both with and without glycans. The average accessible area of protein atoms for non-glycosylated and glycosylated Spike was 1395 nm² and 864 nm², respectively, indicating that glycans shield about 38% of the protein surface of Spike, a value that is comparable to what was previously found in simulations of Spike alone (*Sikora et al., 2021*; *Grant et al., 2020*). The area of

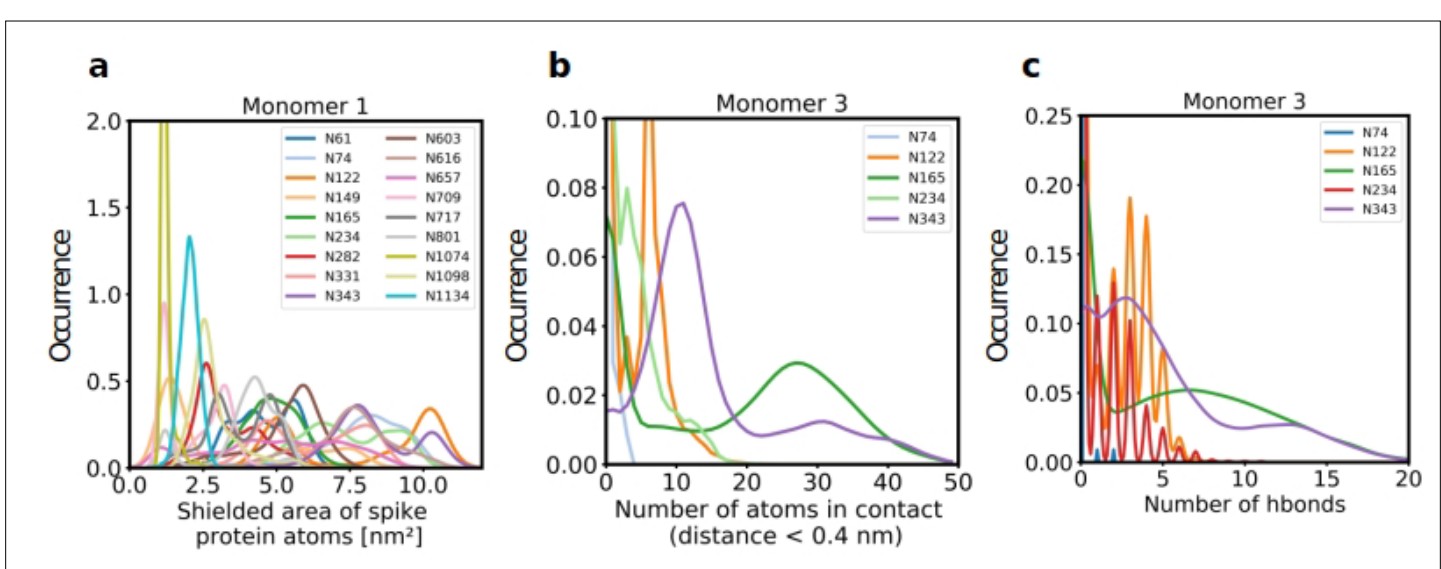

**Figure 2.** Role of Spike glycosylation in shielding the protein surface and interactions with hACE2. (**a**) Normalized distribution of the area of Spike protein atoms that is shielded by each of its glycans on monomer 1 (see *Figure 2—figure supplement 1* for monomers 2 and 3). (**b**) Normalized distribution of the number of atoms in contact with ACE2 and (**c**) the number of hydrogen bonds with ACE2 for glycans on monomer 3 of Spike (see *Figure 2—figure supplement 2* for monomers 1 and 2).

The online version of this article includes the following figure supplement(s) for figure 2:

**Figure supplement 1.** Normalized distribution of the area of Spike atoms shielded by each of its N-glycans.

**Figure supplement 2.** Interactions of Spike N-glycans with ACE2 and with Spike.

protein atoms that are shielded by the individual glycans are shown in *Figure 2a* and *Figure 2—figure supplement 1*.

Further analysis showed that glycans at N122, N165, and N343 on Spike directly interact with ACE2 or its glycans (*Figures 1b, c, 2b and c*). It has been reported that Spike mutants lacking the glycans at N331 and N343 display reduced infectivity, while elimination of the glycosylation motif at N234 results in increased resistance to neutralizing antibodies, without reducing infectivity of the virus (*Li et al., 2020*). The equilibrium between the 'up' and 'down' conformations of Spike involves various stabilizing and destabilizing effects, with possible roles for the glycans at N165, N234, N331, and N343 (*Casalino et al., 2020*; *Sztain et al., 2021*; *Mori et al., 2021*). Removing the glycans at N165, N234 and N343 was experimentally seen to reduce binding to ACE2 by 10%, 40%, and 56%, respectively (*Casalino et al., 2020*; *Sztain et al., 2021*). In our MD simulations, the glycan at position N343 interacts directly with ACE2 (*Figure 2*), while the glycan at N331 interacts with a neighboring Spike monomer (*Figure 1c*, *Figure 2—figure supplement 2*), indicating that the N331 glycosylation site only indirectly affects the interaction of Spike with ACE2. In our model, the glycan at N234 also does not interact directly with ACE2, but seems to stabilize the 'up' conformation. Its removal could favor the 'down' conformation of the RBD, possibly explaining the observed more effective shielding against neutralizing antibodies. In agreement with previous simulations (*Casalino et al., 2020*) the Man9 glycan at N234 of Spike partially inserts itself into the vacant space in the core of the trimer that is created when the RBD of monomer three is in the 'up' conformation (*Figure 1—figure supplement 4*). In our simulations, the free space created by the 'up' conformation seems slightly smaller for Spike in complex with ACE2, suggesting that binding to ACE2 has a stabilizing effect on the Spike monomer.

The N165Q mutant was experimentally found to be more sensitive to neutralization (*Li et al., 2020*). In our models, the glycan at N165 is positioned directly next to the RBD (*Figure 1b*) and thus could shield important antigenic sites. These data highlight the complex impact of Spike glycosylation on the intramolecular interactions of the Spike monomers and, critically, the interaction with ACE2, posing a challenge to design SARS-CoV-2 neutralizing moieties.

Since our modeling clearly confirmed that ACE2 glycosylation plays a significant role in its binding to Spike (*Figure 1d and e*), we also determined the area of the Spike-ACE2 interface region, by subtracting the SASA of the complex from the SASA of the individual proteins and dividing by two. The total interface area was 24.6 nm², with glycans accounting for up to 51% of the interface area, that

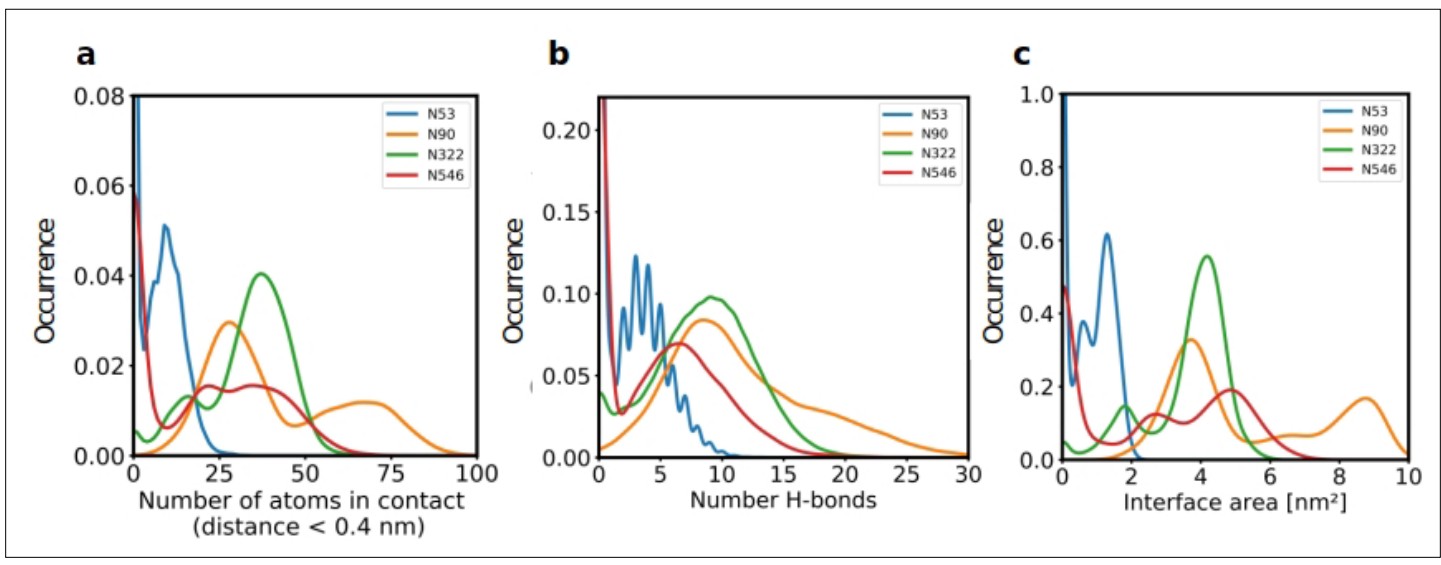

**Figure 3.** The role of hACE2 glycosylation in the interaction with Spike. (**a**) Normalized distribution of the number of atoms of glycans at N53, N90, N322, and N546 of ACE2 that are in contact with Spike (distance <0.4 nm). (**b**) Normalized distribution of the number of hydrogen bonds between glycans at N53, N90, N322, N546 of ACE2 and Spike. (**c**) Normalized distribution of the interface area between Spike and glycans at N53, N90, N322, and N546 of ACE2.

The online version of this article includes the following figure supplement(s) for figure 3:

**Figure supplement 1.** Analysis of the individual MD simulations with separate analysis of the first and second half of the simulations.

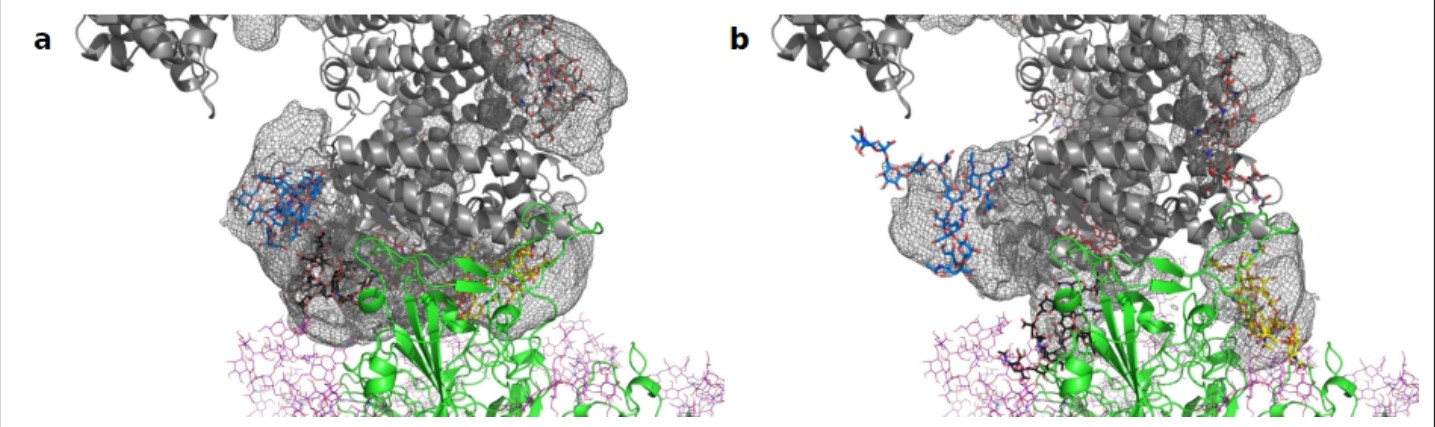

**Figure 4.** Average location density maps of glycans on ACE2. (**a**) The density map (gray mesh) of the glycans at N53, N90, N322, and N546 as observed in the simulations of unbound ACE2 are superimposed onto the ACE2 – Spike complex. (**b**) The density map of the same glycans, as observed in the simulation of the ACE2 – Spike complex. ACE2 in gray, Spike in green. Single, randomly selected conformations of the glycans are shown in blue (N53), yellow (N90), black (N322), and red (N546).

The online version of this article includes the following figure supplement(s) for figure 4:

**Figure supplement 1.** Average location density maps of sialic acids on ACE2 glycans.

is 12.6 nm², contributed by the four most relevant glycans at positions N53, N90, N322, and N546 of ACE2 (*Figure 3*). Furthermore, we scored the number of atoms of each ACE2 glycan in contact with Spike. A contact was defined as a distance of less than 0.4 nm between two atoms. This allowed us to identify the glycans at N53, N90, N322, and N546 as interacting with Spike, with the glycan at position N53 having the weakest interaction. Notably, N546 interacted with Spike for a significant amount of time only in one of the two independent simulations. See *Figure 3—figure supplement 1* for an analysis of the individual MD simulations and for separate analysis of the first and second half of the simulations. Remarkably, the interactions of the glycan at N90 seem to be more pronounced in the second, as compared to the first simulation. The degree of interaction correlated with the spatial proximity between the glycans and the RBD (*Figure 1e and f*). Assessing the number of hydrogen bonds that formed during the simulations, the glycans at N90 and N322 appear most prominent (*Figure 3b*) with a maximum occurrence of 10 hydrogen bonds existing concurrently. Interestingly, the glycans at N90 and N322 interact directly with Spike protein atoms, while the glycan at N546 (red sticks in *Figure 1f*) interacts with the glycans at N122 and N165 of Spike (dark green and orange sticks in *Figure 1b*). The glycans are highly dynamic, as can be seen in the breadth of the distributions of hydrogen bond occurrences and the multimodal character of the distributions for the interface area between the glycans and Spike (*Figure 3c*). A single glycan at N90 is observed to form up to 30 hydrogen bonds to Spike and to be responsible for an interface area of up to 10 nm², representing almost 40% of the average total interface area between Spike and ACE2. These findings are in agreement with previously reported simulations of the complexes (*Zhao et al., 2020*; *Mehdipour and Hummer, 2021*) but raise the question if the glycans contribute favorably to the binding of the two proteins by mediating relevant interactions or unfavorably because of steric restraints and a loss of conformational freedom upon binding.

Next, we assessed the conformational freedom of ACE2 glycans upon binding to Spike and compared their respective density maps in the simulations of free ACE2, and ACE2 in complex with Spike (*Figure 4*). The density map of the unbound ACE2 (*Figure 4a*) shows a continuous density of glycans, largely covering the interface area. Formation of the ACE2-Spike complex significantly reduces the conformational freedom of the glycans, in particular the ones at N90 and N322 (*Figure 4b*). We predict that the glycans at N90 and N322 hamper binding to Spike, either sterically or through an entropic penalty upon binding due to a loss of conformational freedom. These glycans have been implicated as being relevant for binding before (*Zhao et al., 2020*), as well as the glycan at N53 (*Barros et al., 2021*), but no conclusions were drawn if they contribute positively or negatively to binding. Mehdipour and Hummer predicted the glycan at N322 to contribute favorably to binding, because of the favorable interactions of this glycan with the Spike surface (*Mehdipour and Hummer,*

*2021*). We did not observe a significantly more pronounced interaction with Spike for the glycan at N322, compared to the one at N90 (*Figure 3*). Based on conformational considerations, we therefore rather predict a negative impact on binding for both glycans (*Figure 4*).

Since only the glycans at N90 and N322 directly interact with the protein atoms of the Spike proteins, while the glycan on N546 forms hydrogen bonds with glycans present on Spike, we set out to confirm the negative influence of N90 and N322 glycosylation on the interactions with Spike experimentally. First, we ablated N-glycosylation at N90 and N322 individually using the ACE2-Fc fusion constructs ACE2-T92Q-Fc (*Chan et al., 2020*) and ACE2-N322Q-Fc. Note that (*Chan et al., 2020*) indeed suggests that removal of the glycan at N90 through a mutation of T92 leads to enhanced interaction with Spike. The same data set, however, suggests that removal of the glycan at N322 through a mutation of T324 most likely leads to reduced affinity to Spike. However, T324 is itself part of the interface with Spike (*Figure 1—figure supplement 5*), and any mutation of this residue could easily disrupt ACE2 – Spike binding directly, rather than through its effect on the N322 glycosite. We therefore decided to mutate N322 into glutamine to prevent glycosylation at this position.

The wild-type and mutant ACE2-Fc constructs were expressed in HEK293-6E cells and purified from the culture supernatants by protein A affinity chromatography to apparent homogeneity (*Figure 5—figure supplement 1*). Analysis by size-exclusion chromatography combined with detection by multi-angle light scattering (SEC-MALS) demonstrated that all purified proteins were dimers of the expected native molecular mass (*Figure 5—figure supplement 2*). The impact of the introduced mutations on the overall fold of ACE2-Fc was tested with differential scanning calorimetry (DSC), a sensitive biophysical method for the assessment of the thermal stability of proteins. Three thermal transitions could be discriminated. The first midpoint of transition ($T_m1$) is due to the unfolding of ACE2, whereas the second and third midpoints of transitions ($T_m2$ and $T_m3$) reflect the thermal denaturation of the $C_H2$ and $C_H3$ domains of the Fc part of the fusion proteins (*Lobner et al., 2017*). The $T_m1$ midpoint transition temperatures of the ACE2-Fc glycomutants (53.3°C–54.0°C) were slightly higher than for the wild-type protein (52.2 °C), while $T_m2$ and $T_m3$ remained unchanged (*Figure 5*). This indicates that removal of the N90 and N322 glycans does not compromise the structural integrity of ACE2.

The Spike-binding properties of the purified ACE2-Fc variants were characterized by biolayer interferometry (BLI). For this, ACE2-wt-Fc, ACE2-T92Q-Fc, and ACE2-N322Q-Fc were biotinylated, immobilized on streptavidin biosensor tips and dipped into serial dilutions of trimeric Spike. Since we did not observe appreciable dissociation of ACE2-Fc/trimeric Spike complexes in our analyses (*Figure 6—figure supplement 1*), we evaluated the association rates ($k_{obs}$; *Figure 6a*). To determine equilibrium affinity constants ($K_D$), we analyzed the interactions between the immobilized ACE2-Fc constructs and monomeric RBD (*Figure 6b*, *Figure 6—figure supplement 2*). The BLI data are in good agreement with our computational models, confirming that the removal of protein N-glycosylation at either N90 or N322 results in up to twofold higher binding affinities, when compared to ACE2-wt-Fc (ACE2-wt-Fc: $K_D$ = 16.2 ± 0.7 nM; ACE2-T92Q-Fc: $K_D$ = 8.0 ± 0.7 nM; ACE2-N322Q-Fc: $K_D$ = 11.4 ± 0.3 nM; *Figure 6b*; *Figure 6—figure supplement 2*). Thus, structure-guided glyco-engineering at N90 and N322 results in ACE2 forms with increased affinity for SARS-CoV-2 Spike binding.

Next, we tested the virus neutralization properties of ACE2-wt-Fc, ACE2-T92Q-Fc, and ACE2-N322Q-Fc. For this, we infected Vero E6 cells with 60 plaque-forming units (PFU; multiplicity of infection (MOI): 0.002) of SARS-CoV-2 in the presence of 10–50 µg/mL ACE2-wt-Fc, ACE2-T92Q-Fc, or ACE2-N322Q-Fc. The extent of SARS-CoV-2 infection and replication was quantified by RT-qPCR detection of viral RNA present in the culture supernatants. Untreated SARS-CoV-2 infected cells released up to 10 times more viral RNA than ACE2-wt-Fc-treated cells. Importantly, co-incubation of cells with SARS-CoV-2 and ACE2-T92Q-Fc resulted in significant further reduction of the viral load when compared to ACE2-wt-Fc. Enhanced SARS-CoV-2 neutralization was also observed for ACE2-N322Q-Fc. However, this mutant was less effective in promoting virus neutralization than ACE2-T92Q-Fc (*Figure 7* and *Figure 7—figure supplement 1*). Similar results were obtained when SARS-CoV-2 neutralization assays were performed with much larger amounts of inoculated virus (MOI: 20) and concomitantly increased ACE2-Fc concentrations (*Figure 7—figure supplement 2*). Hence, in line with our structural glycan interaction map, the removal of either of the N-glycans attached to N90 and N322 gives rise to ACE2 decoy receptors with improved SARS-CoV-2 neutralization properties.

To investigate a potential additive effect of simultaneous elimination of N-glycosylation at N90 and N322, we generated a double mutant ACE2-T92Q-N322Q-Fc construct. We also digested

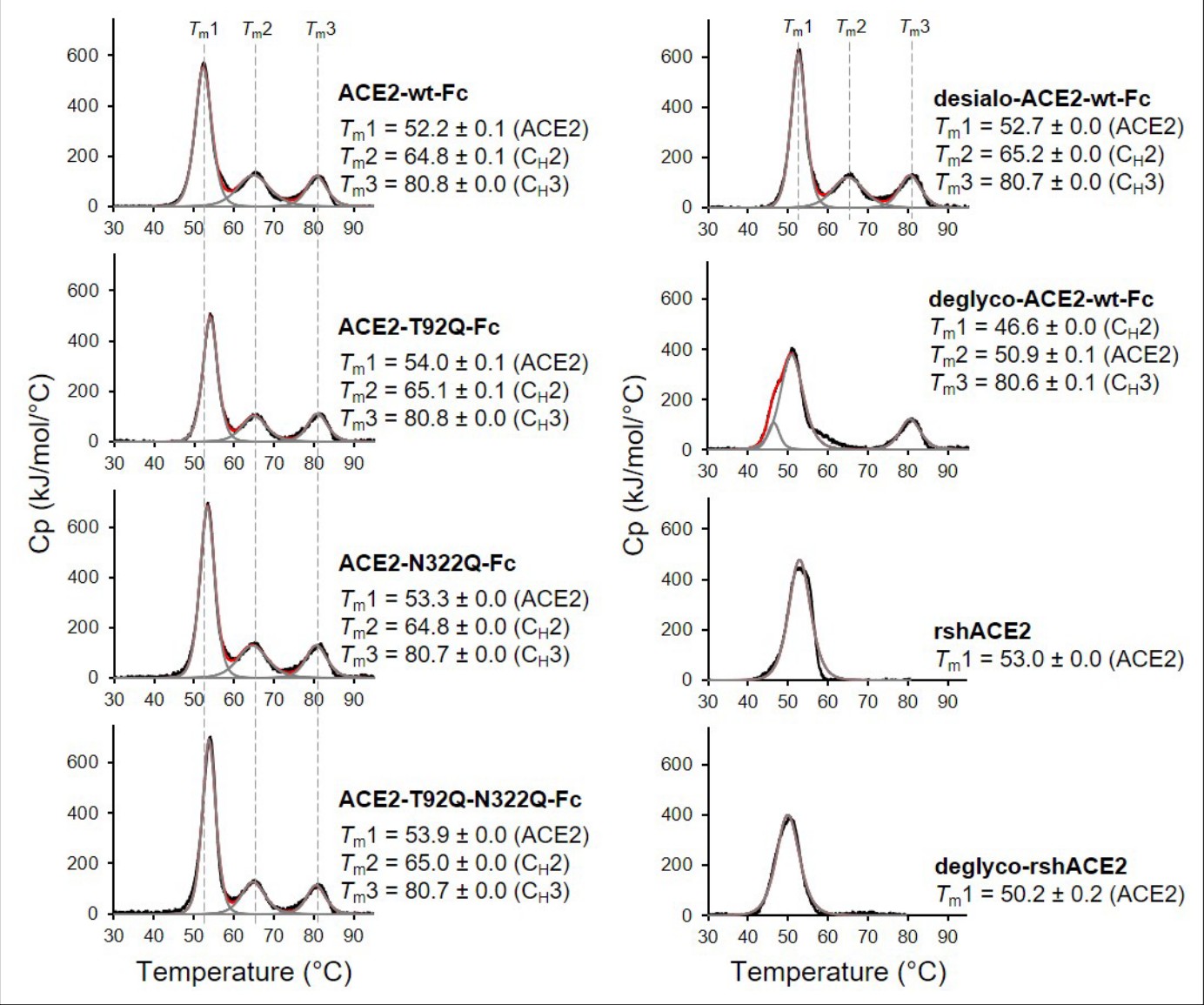

**Figure 5.** Analysis of ACE2 variants by differential scanning calorimetry (DSC). Raw data (black) were smoothened (red) and then fitted using a non-two-state thermal unfolding model (gray). Data are presented as mean ± SEM of three independent experiments. Cp, heat capacitance; rshACE2, clinical-grade recombinant soluble human ACE2; deglyco-rshACE2, enzymatically deglycosylated rshACE2; deglyco-ACE2-wt-Fc, enzymatically deglycosylated wild-type ACE2-Fc; desialo-ACE2-wt-Fc, enzymatically desialylated wild-type ACE2-Fc.

The online version of this article includes the following figure supplement(s) for figure 5:

**Figure supplement 1.** Analysis of ACE2 and Spike variants by SDS-PAGE.

**Figure supplement 2.** Analysis of ACE2 and Spike variants by SEC-MALS.

ACE2-wt-Fc with peptide-N4-(N-acetyl-beta-glucosaminyl)asparagine amidase F (PNGase F) to remove all accessible N-glycans (deglyco-ACE2-wt-Fc) and neuraminidase to release terminal sialic acid residues (desialo-ACE2-wt-Fc). Purity and homogeneity of these additional ACE2-Fc variants was ascertained by SDS-PAGE and SEC-MALS (*Figure 5—figure supplements 1 and 2*). The absence of N-glycans attached to N90 and/or N322 in ACE2-T92Q-N322Q-Fc and the respective single mutants was demonstrated by LC-ESI-MS (*Figure 7—figure supplement 3*). Quantitative release of sialic acids and complete removal of N-glycans from all ACE2-wt-Fc N-glycosylation sites with the exception of N546 was also confirmed (*Figure 7—figure supplements 4 and 5*). The glycans at N546 of ACE2-wt-Fc exhibited partial resistance (40%) to PNGase F treatment (*Figure 7—figure supplement 4*). Combined introduction of the mutations T92Q and N322Q as well as enzymatic desialylation did

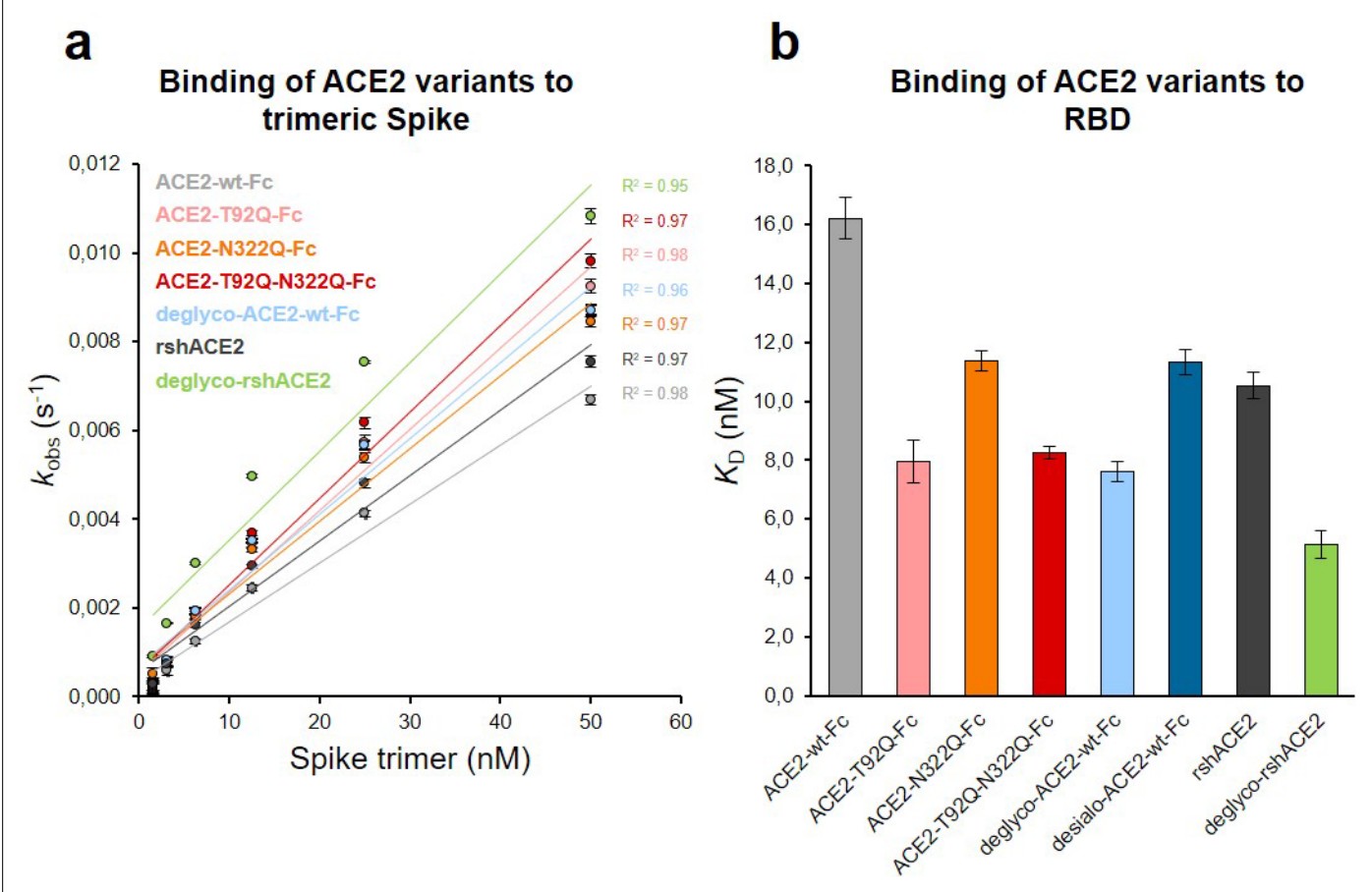

**Figure 6.** Binding of Spike and RBD to glyco-engineered ACE2 variants. (**a**) Binding of Spike to glyco-engineered ACE2 variants as determined by biolayer interferometry (BLI). Plots of $k_{obs}$ (observed association rate) as a function of Spike concentration were generated by fitting the association data to a 1:1 binding model. Binding analysis was performed by dipping ACE2-loaded biosensors into twofold serial dilutions of purified Spike (1.6–50 nM). All measurements were performed in triplicates. rshACE2, clinical-grade recombinant soluble human ACE2; deglyco-rshACE2, enzymatically deglycosylated rshACE2; deglyco-ACE2-wt-Fc, enzymatically deglycosylated wild-type ACE2-Fc. (**b**) $K_D$ values for the interaction of the indicated glyco-engineered ACE2 variants with monomeric RBD. Data are presented as mean ± SEM of 3 independent experiments. Desialo-ACE2-wt-Fc, enzymatically desialylated wild-type ACE2-Fc.

The online version of this article includes the following figure supplement(s) for figure 6:

**Figure supplement 1.** Binding of Spike to human ACE2 variants as determined by biolayer interferometry.

**Figure supplement 2.** Binding kinetics of RBD to human ACE2 variants as determined by biolayer interferometry.

not reduce the thermal stability of ACE2-Fc as assessed by DSC, while close-to-complete removal of N-glycans by PNGase F led to a slightly decreased midpoint transition temperature of the ACE2 domain (*Figure 5*). Studies of the interaction between ACE2-T92Q-N322Q-Fc and deglyco-ACE2-wt-Fc with RBD by BLI analysis yielded $K_D$ values similar to those determined for the single mutant ACE2-T92Q-Fc (ACE2-T92Q-N322Q-Fc: $K_D$ = 8.2 ± 0.2 nM; deglyco-ACE2-wt-Fc: $K_D$ = 7.6 ± 0.3 nM). The affinity of desialo-ACE2-wt-Fc for RBD ($K_D$ = 11.3 ± 0.4 nM) was also higher than that of native ACE2-wt-Fc (*Figure 6b*). The increased affinities of these ACE2-Fc variants for Spike correlate with their potencies to neutralize SARS-CoV-2, with deglyco-ACE2-wt-Fc followed by ACE2-T92Q-N322Q-Fc displaying the highest neutralization potencies (*Figure 7* and *Figure 7—figure supplement 2*). The effect of desialo-ACE2-wt-Fc on SARS-CoV-2 infections of Vero E6 cells was less pronounced and comparable to that of the single mutant ACE2-N322Q-Fc (*Figure 7*), in good agreement with the almost identical RBD-binding affinities of these two ACE2-Fc variants (*Figure 6b*). Taken together, these data identify critical glycans at position N90 and N322 of ACE2 that structurally and functionally interfere with Spike-ACE2 binding; ablation of these glycans via site-directed mutagenesis

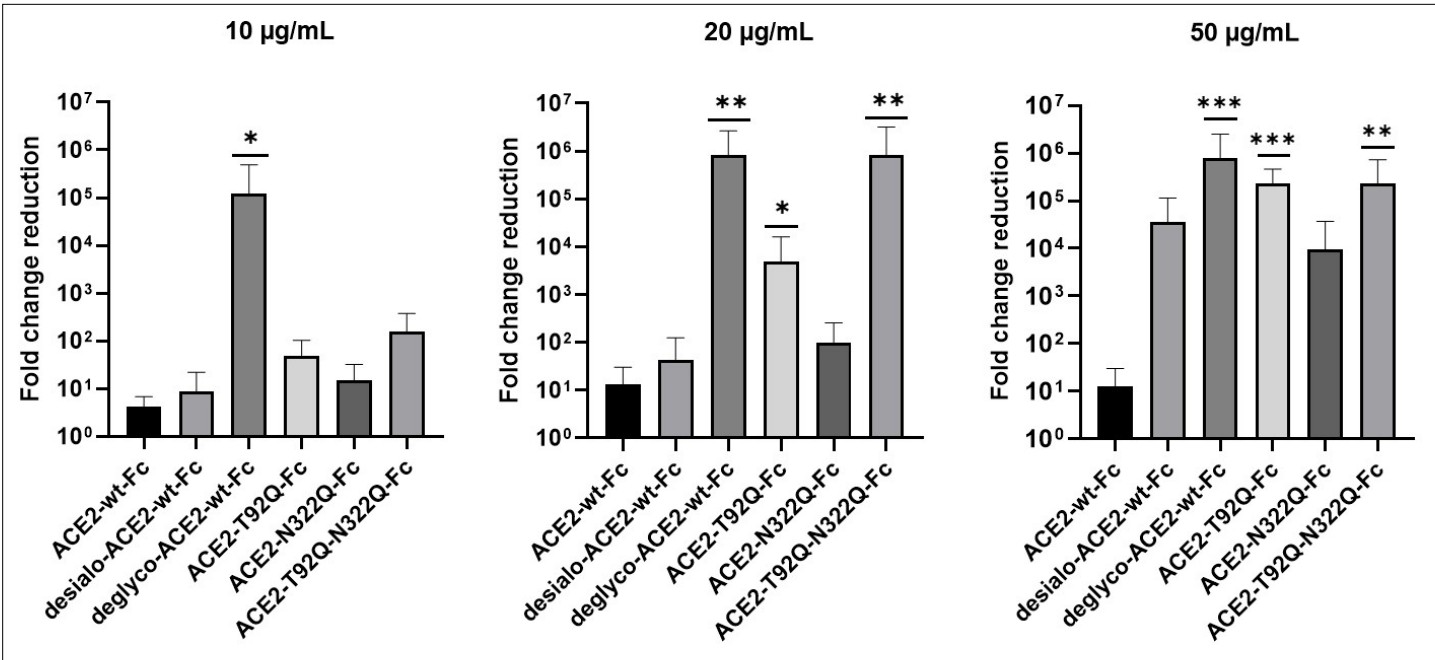

**Figure 7.** Critical role of ACE2 glycosylation for SARS-CoV-2 infectivity. Inhibition of SARS-CoV-2 infection of Vero E6 cells using wild-type ACE2-Fc and the indicated glyco-engineered ACE2-Fc variants at final concentrations of 10–50 µg/mL. The viral RNA content of the culture supernatants was quantified by RT-qPCR and expressed as fold change reduction relative to untreated controls. With the exception of ACE2-T92Q-Fc, all data are presented as mean ± SEM of three independent experiments each performed in triplicates. In the case of ACE2-T92Q-Fc, only two independent experiments could be performed due to the limited availability of this protein. Deglyco-ACE2-wt-Fc, enzymatically deglycosylated wild-type ACE2-Fc; desialo-ACE2-wt-Fc, desialylated wild-type ACE2-Fc. $*p < 0.05$; $**p < 0.01$; $***p < 0.001$ (Kruskal-Wallis).

The online version of this article includes the following figure supplement(s) for figure 7:

**Figure supplement 1.** Critical role of ACE2 glycosylation for SARS-CoV-2 infectivity.

**Figure supplement 2.** Critical role of ACE2 glycosylation for SARS-CoV-2 infectivity.

**Figure supplement 3.** Site-specific ablation of ACE2 N-glycans by mutagenesis.

**Figure supplement 4.** Deglycosylation of ACE2-wt-Fc and rshACE2 by PNGase F.

**Figure supplement 5.** Analysis of enzymatically desialylated ACE2-wt-Fc.

or enzymatic deglycosylation generated ACE2 variants with improved Spike-binding properties and increased neutralization strength.

The results presented above uncover the critical importance of N-glycans located at the ACE2-Spike interface for the infection of host cells by SARS-CoV-2. This prompted us to test the feasibility of removing all N-glycans from clinical-grade rshACE2, which has undergone placebo-controlled phase II clinical testing in 178 COVID-19 patients (ClinicalTrials.gov Identifier: NCT04335136), and to test for its SARS-CoV-2 neutralization properties. To this end, we generated enzymatically deglycosylated clinical-grade rshACE2 (deglyco-rshACE2) using PNGase F. The quantitative release of all N-glycans, with the exceptions of those attached to the N432 and N546 glycosites, was confirmed by LC-ESI-MS/MS (*Figure 7—figure supplement 4*), and the integrity and homogeneity of dimeric deglyco-rshACE2 was demonstrated by SEC-MALS (*Figure 5—figure supplement 2*). Paralleling our observations with deglyco-ACE2-wt-Fc, we found the binding affinity of deglyco-rshACE2 to RBD ($K_D = 5.1 \pm 0.5$ nM) to be two times higher than for native rshACE2 ($K_D = 10.5 \pm 0.4$ nM; *Figure 6b*). Furthermore, deglyco-rshACE2 displayed improved SARS-CoV-2 neutralization properties in Vero E6 cell infection assays. At a final concentration of 200 µg/mL deglyco-rshACE2, we observed a significant reduction in SARS-CoV-2 replication when compared to treatment with the native form of the protein (*Figure 8*).

Besides serving as a soluble decoy receptor to prevent SARS-CoV-2 infection of ACE2-expressing host cells, rshACE2 also regulates blood pressure and protects multiple organs such as the heart, kidney and lung as well as blood vessels via enzymatic degradation of angiotensin II (*Vickers et al., 2002*). In contrast to other recently described ACE2 mutants displaying improved Spike binding

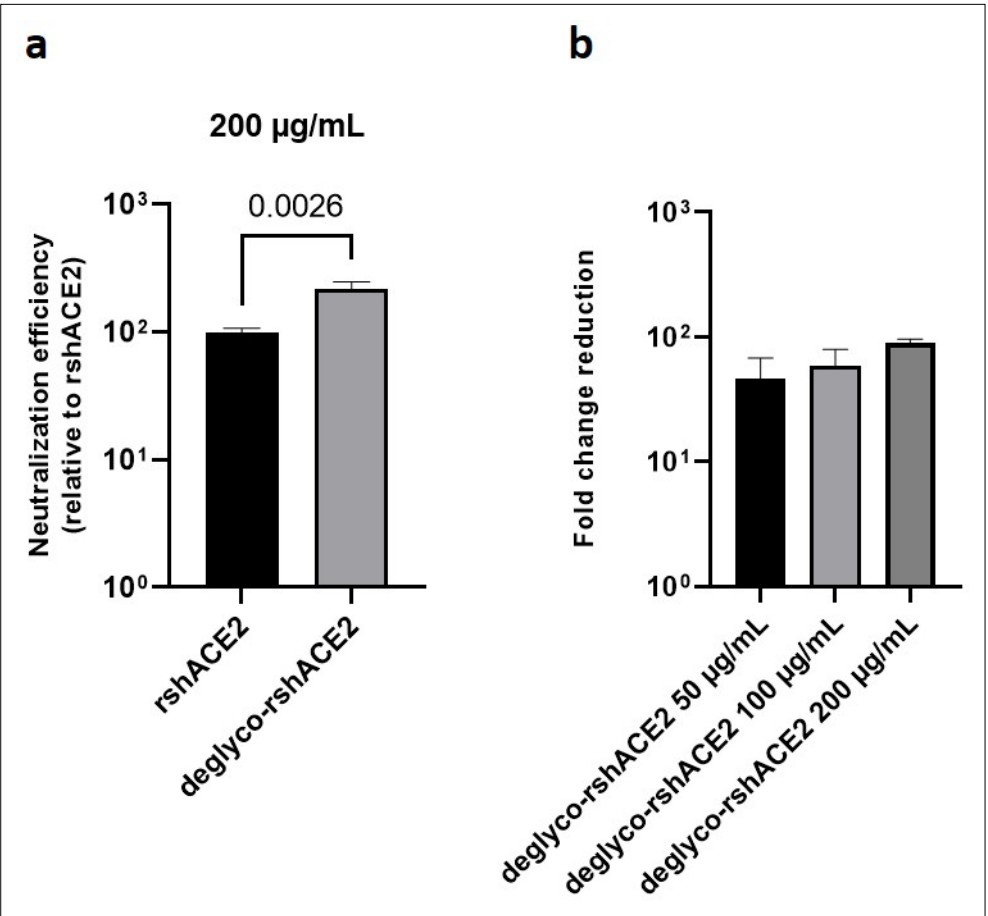

**Figure 8.** Deglycosylated rshACE2 is a potent SARS-CoV-2 decoy receptor. (**a**) Inhibition of SARS-CoV-2 infection of Vero E6 cells at an MOI of 20 using native and enzymatically deglycosylated rshACE2 at final concentrations of 200 µg/mL. The viral RNA content of the infected cells was quantified by RT-qPCR and expressed as neutralization efficiency relative to native rshACE2 (set to 100%). Data are presented as mean ± SEM of four independent experiments. Deglyco-rshACE2, enzymatically deglycosylated rshACE2; p = 0.0026 (Student's *t*-test). (**b**) Inhibition of SARS-CoV-2 infection of Vero E6 cells at an MOI of 20 using deglyco-rshACE2 at final concentrations of 50–200 µg/mL. The viral RNA content of the infected cells was quantified by RT-qPCR and expressed as fold change reduction relative to untreated controls. Data are presented as mean ± SD of triplicates.

concomitant with inadvertently or intentionally impaired enzymatic activity (*Glasgow et al., 2020*; *Chan et al., 2020*), the catalytic activities of ACE2-T92Q-Fc, ACE2-N322Q-Fc and ACE2-T92Q-N322Q-Fc were found to be only modestly reduced as compared to ACE2-wt-Fc (ACE2-T92Q-Fc: 65% ± 11%; ACE2-N322Q-Fc: 69% ± 7%; ACE2-T92Q-N322Q-Fc: 79% ± 11%; *Figure 9*).

Interestingly, deglyco-ACE2-wt-Fc (149% ± 1 %) and desialo-ACE2-wt-Fc (160% ± 2 %) exhibited higher enzymatic activities than native ACE2-wt-Fc (*Figure 10* and *Figure 10—figure supplement 1*). A similar observation was made for deglyco-rshACE2, although the enhancing effects of enzymatic deglycosylation on catalytic efficiency were less pronounced (113% ± 2% as compared to native rshACE2; *Figure 10*). These results show that enzymatic removal of N-glycans from ACE2-Fc and clinical-grade rshACE2 results in increased Spike binding and enhanced SARS-CoV-2 neutralization while preserving its potentially critical enzymatic activity.

## Discussion

Our data demonstrate that structure-guided glycoengineering is a powerful means to develop ACE2 variants with improved SARS-CoV-2 neutralization properties without compromising the structural stability and catalytic activity of the enzyme. Our in silico models of the Spike-ACE2 complex combined

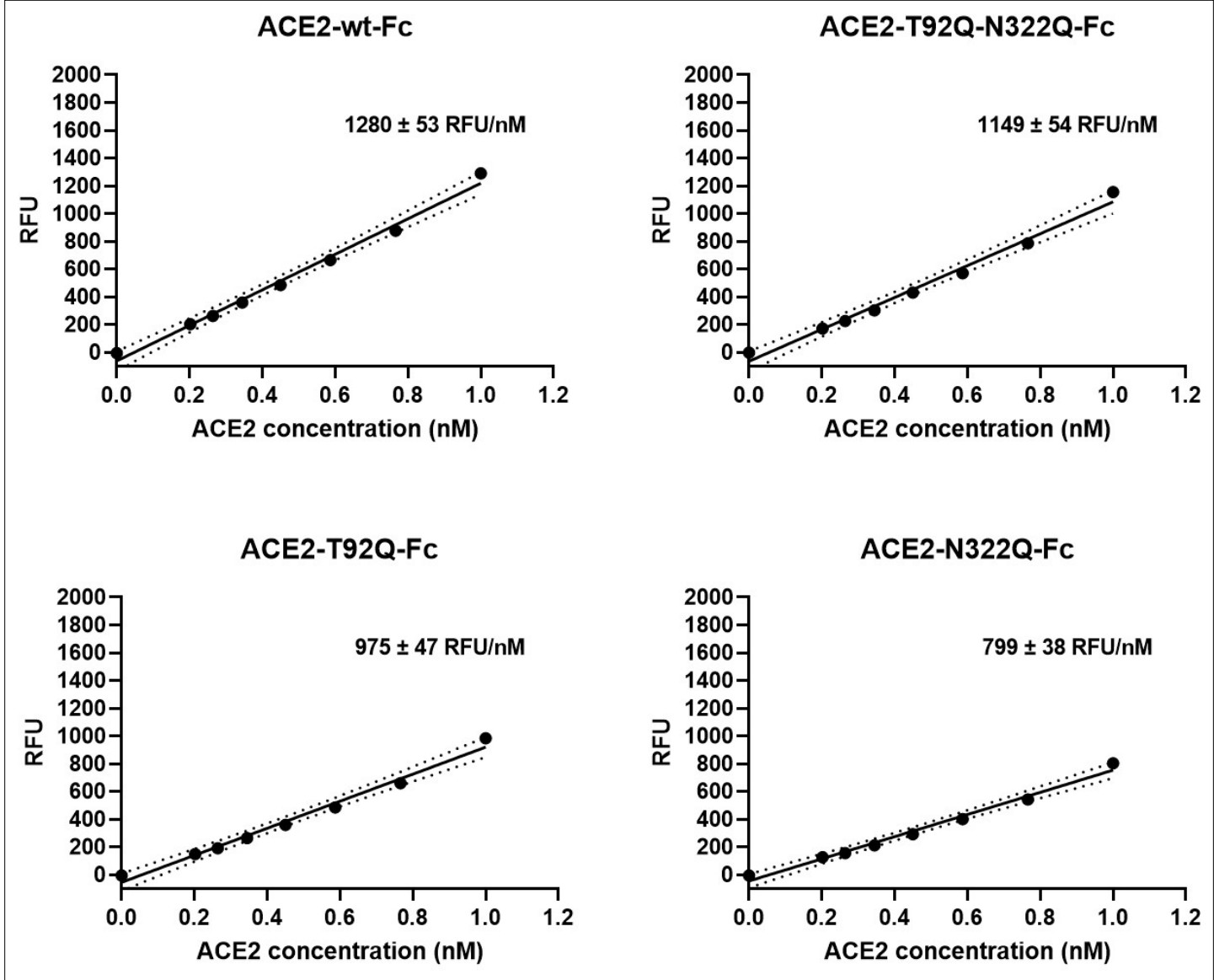

**Figure 9.** Enzymatic activity of ACE2-Fc mutants. Hydrolysis of 100 μM 7-methoxycoumarin-4-yl-acetyl-Ala-Pro-Lys-2,4-dinitrophenyl was continuously monitored by spectrofluorimetry. Hydrolytic activity is plotted as relative fluorescence units (RFU) over ACE2-Fc concentration (in nM). All assays were performed in technical triplicates. One representative experiment out of two is shown.

with simulations of its spatial and temporal dynamics rationalized previously published data and led to predictions that were confirmed by in vitro binding studies and cell-based SARS-CoV-2 neutralization assays. However, it requires further explanation how the moderately enhanced affinity of ACE2 glyco-variants for monomeric RBD observed in biolayer interferometry experiments can relate to a far more pronounced increase of their virus-neutralization potency in Vero E6 cells. First, while the binding affinity is represented by an equilibrium constant, which is the ACE2 concentration at which 50% is in a bound state, the inhibitory strength in cell-based assays is measured at three distinct concentrations, at which one protein may show little inhibition (the relative concentration in the assay is below the $K_d$) while another variant shows strong inhibition (the relative concentration in the assay is above the $K_d$). Second, a cooperative effect may be expected for the association of trimeric Spike molecules present in the viral envelope with ACE2 dimers. In this supramolecular setting, a subtle increase in the affinity of ACE2 for RBD can lead to a dynamic equilibrium of binding and unbinding events with up to six potential interactions, leading to an overall much stronger avidity effect. The dynamic equilibrium of the RBDs between 'down' and 'up' conformations can further steepen the dose-response

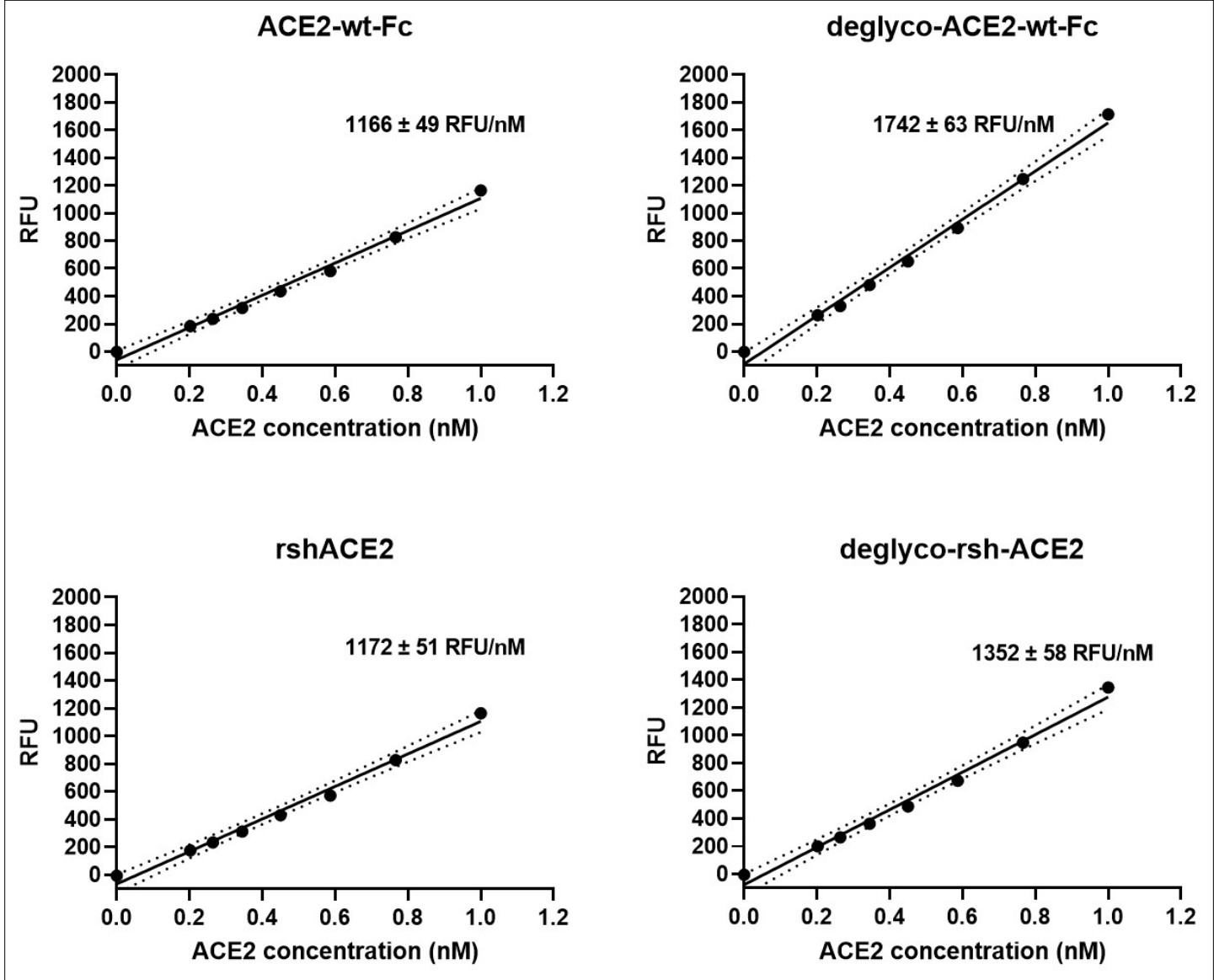

**Figure 10.** Enzymatic activity of enzymatically deglycosylated ACE2. Hydrolysis of 100 µM 7-methoxycoumarin-4-yl-acetyl-Ala-Pro-Lys-2,4-dinitrophenyl was continuously monitored by spectrofluorimetry. Hydrolytic activity is plotted as relative fluorescence units (RFU) over ACE2 concentration (in nM). All assays were performed in technical triplicates. One representative experiment out of two is shown.

The online version of this article includes the following figure supplement(s) for figure 10:

**Figure supplement 1.** Enzymatic activity of desialo-ACE2-wt-Fc.

curves (*Monod et al., 1965*). Third, a slight advantage of the soluble ACE2 decoy receptor over endogenous native ACE2 may be sufficient to tip the balance between SARS-CoV-2 attachment and shedding of viral particles from the host cell surface. The inhibitory strength is not a function of the number of Spike proteins that are bound, but of the probability that a high enough fraction of Spike proteins is bound, potentiating a small increase in affinity (*Magnus, 2013*). Fourth, the SARS-CoV-2 neutralization assays shown in *Figure 7* were performed at a low multiplicity of infection. This can lead to complete neutralization of all virus particles in the inoculum (*Monteil et al., 2020*) as becomes evident from the individual data points of our experiments (*Figure 7—figure supplement 1*). Hence, the data are presented as medians and not as arithmetic means, which potentially accentuates the numeric differences between individual samples. Finally, it is possible that the N-glycan moiety of ACE2 also modulates other aspects of viral entry besides promoting the docking of Spike to the cell surface (*Yang et al., 2020*).

It has been reported that the sialylation status of ACE2 affects its interactions with SARS-CoV-2 Spike (*Allen et al., 2021*). We have found that enzymatic desialylation of ACE2 results in a reproducible increase of its affinity to RBD without detectable structural penalties. Importantly, desialylated ACE2 is more efficient in neutralizing SARS-CoV-2 than its native counterpart. Molecular simulations suggest that the terminal sialic acids of the N-glycans attached to ACE2 residues N90 and N322 mask parts of the Spike-ACE2 interface and thus could interfere with Spike binding through steric clashes and/or electrostatic effects (*Figure 4—figure supplement 1*). This provides a structural rationale how sialic acids present on ACE2 might dampen interactions with Spike during SARS-CoV-2 attachment to host cells (*Chu et al., 2021*).

We want to point out that all virus-neutralization experiments presented in this paper were performed using Vero E6 cells. Although this cell line is very popular for SARS-CoV-2 propagation and infectivity studies, it will be important to repeat our neutralization assays with a cell line derived from human lung (e.g. Calu-3) (*Johnson et al., 2021*). Furthermore, treatment with deglycosylated ACE2 could have per se a negative impact on cell viability and thus affect virus replication in the treated cultures. We will assess the potential cellular toxicity of deglycosylated ACE2 in future studies. However, we have shown previously that native ACE2 does not display any cytotoxic effects even when used at very high concentrations (*Monteil et al., 2020*).

In line with other reports (*Chan et al., 2020*; *Chu et al., 2021*), our results indicate that the elimination of the N-glycans attached to N90 is largely responsible for the improved Spike-binding properties of enzymatically deglycosylated ACE2. As proposed (*Glasgow et al., 2020*; *Chan et al., 2020*) and corroborated by our mutational analysis, substitution of ACE2 residues N90 or T92 could indeed provide an alternative approach for the development of ACE2 variants with improved SARS-CoV-2 sequestering properties. Our data indicate that ablation of N90 glycosylation could be combined with mutations of N322 and possibly other ACE2 N-glycosylation sites to achieve an even higher SARS-CoV-2 neutralizing potency. However, expression of an ACE2 variant lacking all potential N-glycosylation sites in ACE2-negative host cells led to reduced rather than enhanced susceptibility of the cells to SARS-CoV-2 as compared to transduction with wild-type ACE2 (*Chu et al., 2021*). This was attributed to the much lower cellular content of the mutant protein relative to the native enzyme, thus demonstrating that the importance of N-glycosylation for proper folding of glycoproteins during their biosynthesis (*Xu and Ng, 2015*) also applies to ACE2. Given the inferior expression yields of glycan-free ACE2 and the potential of unwanted immunological side effects when non-natural mutations are introduced into a therapeutic glycoprotein, we believe that the clinical potential of enzymatically deglycosylated rshACE2 is superior to that of any of our ACE2 glycomutants. In our opinion, treatment of clinical-grade rshACE2 with deglycosylation enzymes such as PNGase F followed by a final polishing step represents a straightforward, Good Manufacturing Practice (GMP)-compliant and industrially feasible alternative to generate a potent therapeutic drug for the treatment of SARS-CoV-2 infected persons and patients.

## Materials and methods

**Key resources table**

| Reagent type (species) or resource | Designation | Source or reference | Identifiers | Additional information |
|---|---|---|---|---|
| Cell line (*Homo sapiens*) | HEK293-6E | National Research Council of Canada | RRID: CVCL_HF20 | *Durocher et al., 2002* |
| Cell line (*Cercopithecus aethiops*) | Vero E6 | ATCC | CRL-1586; RRID: CVCL_0574 | also known as VERO C1008 |
| Recombinant DNA reagent | PCAGGS-RBD (plasmid) | Florian Krammer, Icahn School of Medicine at Mount Sinai | | *Amanat et al., 2020* |
| Recombinant DNA reagent | pCAGGS-Spike (plasmid) | Florian Krammer, Icahn School of Medicine at Mount Sinai | | *Amanat et al., 2020* |

*Continued on next page*

*Continued*

| Reagent type (species) or resource | Designation | Source or reference | Identifiers | Additional information |
|---|---|---|---|---|
| Recombinant DNA reagent | pcDNA3-sACE2(WT)-Fc(IgG1) (plasmid) | Addgene | Cat #: 145163; RRID: Addgene_145163 | *Chan et al., 2020* |
| Recombinant DNA reagent | pcDNA3-sACE2-T92Q-Fc(IgG1) (plasmid) | Addgene | Cat #: 145170; RRID: Addgene_145170 | *Chan et al., 2020* |
| Recombinant DNA reagent | pcDNA3-sACE2-N322Q-Fc(IgG1) (plasmid) | this paper | | |
| Recombinant DNA reagent | pcDNA3-sACE2-T92Q-N322Q-Fc(IgG1) (plasmid) | this paper | | |
| Peptide, recombinant protein | soluble human ACE2 | Apeiron Biologicals, Vienna, Austria | APN01 | *Monteil et al., 2020* |
| Biological sample (*Severe acute respiratory syndrome coronavirus 2*) | SARS-CoV-2 | *Monteil et al., 2020* | | GenBank MT093571 |
| Biological sample (*Severe acute respiratory syndrome coronavirus 2*) | SARS-CoV-2 | Charité, Berlin, Germany | Ref-SKU #: 026 V-03883 | |
| Commercial assay or kit | QuikChange Lightning Site-Directed Mutagenesis kit | Agilent Technologies | Cat #: 210,518 | |
| Commercial assay or kit | EZ-Link Sulfo-NHS-LC-Biotin kit | Thermo Fisher Scientific | Cat #: 21,435 | |
| Commercial assay or kit | QiaAmp Viral RNA Minikit | Qiagen | Cat #: 52,904 | |
| Commercial assay or kit | QuantiTect Multiplex RT-qPCR Kit | Qiagen | Cat #: 204,443 | |
| Sequence-based reagent | 2019-nCoV_N1-F | This paper | qPCR primer | GACCCCAAAATCAGCGAAAT |
| Sequence-based reagent | 2019-nCoV_N1-R | This paper | qPCR primer | TCTGGTTACTGCCAGTTGAATCTG |
| Sequence-based reagent | 2019-nCoV_N1-P | This paper | qPCR probe | FAM-ACCCCGCATTACGTTTGGTGGACC-BHQ1 |
| Other | FreeStyle F17 medium | Thermo Fisher Scientific | Cat #: A1383502 | |
| Other | PNGase F | New England BioLabs | Cat #: P0705 | 180,000 U mL$^{-1}$ |
| Other | Neuraminidase | New England BioLabs | Cat #: P0720 | 2,500 U mL$^{-1}$ |
| Chemical compound, drug | Mca-Ala-Pro-Lys (Dnp)-OH | Bachem, Bubendorf, Switzerland | Cat #: 4042638 | 7-methoxycoumarin-4-yl-acetyl-Ala-Pro-Lys-2, 4-dinitrophenyl |
| Software, algorithm | Gromacs | https://www.gromacs.org/ | 2019.4; RRID: SCR_014565 | |
| Software, algorithm | GROMOS | http://www.gromos.net/ | 1.5.0 | |
| Software, algorithm | Octet data analysis software | ForteBio | 11.1.1.39 | |
| Software, algorithm | ASTRA six software | Wyatt Technology | 6 | |
| Software, algorithm | Origin 7.0 for DSC software | Malvern Panalytical | 7.0 | |
| Software, algorithm | GraphPad Prism 8 | GraphPad Software | 8; RRID: SCR_002798 | |

## Modeling of the Spike-hACE2 complex

To model the fully glycosylated SARS-CoV-2 Spike-human ACE2 (hACE2) complex, a protein model was created using partial experimental structures deposited in the protein databank (PDB). The Spike

RBD domain in complex with hACE2 (*Yan et al., 2020*) (PDB: 6M17) was superimposed with the opened RBD domain in a Spike structure with one open RBD domain (*Walls et al., 2020*) (PDB: 6VYB). Alternative Spike structures have been published, which show very similar conformations (*Wrapp et al., 2020*). Similarly, further structures of the Spike RBD-hACE2 complex *Wang et al., 2020*, *Lan et al., 2020* have been reported which show very similar conformations to the templates used. Missing residues in Spike were modeled using SWISS-MODEL (*Waterhouse et al., 2018*) and the superimposed structure as template based on the complete SARS-CoV-2 S sequence (GenBank QHD43416.1).

Different types of glycans were added to Spike and hACE2. For Spike, the assignments of *Watanabe et al., 2020* were followed, selecting oligomannosidic (Man5 or Man9) or complex (bi-antennary di-sialylated core fucosylated; NaNaF) glycans according to the majority of the glycans detected at the respective site. This was largely confirmed by our own analysis (*Hoffmann et al., 2021*). For hACE2, complex (i.e. bi-antennary di-sialylated core fucosylated) N-glycans were added. See *Supplementary file 1* for the exact assignments. Initial conformations of the glycans were selected following previously derived procedures (*Turupcu and Oostenbrink, 2017*). In brief, molecular dynamics simulations were performed of mini-peptides with the glycans attached. Local Elevation (*Huber et al., 1994*) was used to enhance the sampling of all glycosidic linkages, during simulations of 100 ns. The entire glycan trees were clustered based on the conformations of the individual glycosidic linkages (*Perić-Hassler et al., 2010*). This resulted in conformational bundles containing 1301, 1340, and 2413 distinct conformations of Man5, Man9, and NaNaF, respectively. These conformations were fitted onto the respective glycosylation site in the Spike-hACE2 complex using a superposition of the backbone of the asparagine residues and the non-bonded interaction energy between the glycan and protein atoms or previously added glycans was computed. The lowest energy conformation was retained. Topologies and initial conformations were generated using the gromos ++ suite of pre- and post-MD tools (*Eichenberger et al., 2011*). Glycans were added to the complex sequentially, to avoid collisions between individual glycans. A few modeled glycans were incompatible with loops of the Spike protein not resolved in the experimental structures. Loops involved in these structural incompatibilities (residues 141–165 and 471–490) were partially re-modeled in the fully glycosylated model using the RCD+ loop modeling server (*Chys and Chacón, 2013*; *López-Blanco et al., 2016*). The final model was energy-minimized with the GROMOS 54A8 protein force-field (*Reif et al., 2012*; *Reif et al., 2013*), the GROMOS 53A6glyc glycan force-field (*Turupcu and Oostenbrink, 2017*; *Pol-Fachin et al., 2012*; *Pol-Fachin et al., 2014*) and the GROMOS simulation software using the steepest decent algorithm (*Schmid et al., 2012*).

## Molecular dynamics simulations

Molecular dynamics simulations were performed using the simulation package Gromacs (Version 2019.5) and the indicated force field parameters. hACE2 was reduced to residues 21–730 in the models, to reduce its overall size prior to simulation. The models were placed in rhombic dodecahedron simulation boxes and solvated by explicit SPC water molecules (*Berendsen et al., 1981*). This resulted in simulation systems of $5.9 \times 10^5$ and $2.2 \times 10^6$ atoms for hACE2 and the Spike-hACE2 complex, respectively. Two independent 100-ns molecular dynamics simulations were performed for hACE2 and for the Spike-hACE2 complex each. The equations of motion were integrated using a leapfrog integration scheme (*Hockney, 1970*) with a time-step of 2 fs. Non-bonded interactions were calculated within a cutoff sphere of 1.4 nm and electrostatic interactions were computed using a particle-particle particle-mesh (P3M) approach (*Hockney and Eastwood, 1988*). Bond-lengths were constrained to their optimal values using the Lincs algorithm (*Hess et al., 1997*). Temperature was maintained at a constant value using a velocity-rescaling algorithm (*Berendsen et al., 1984*; *Bussi et al., 2007*) with a relaxation time of 0.1 ps. Pressure was maintained constant using a Parrinello-Rahman barostat (*Parrinello and Rahman, 1981*; *Nosé and Klein, 2006*) with a relaxation time of 2.0 ps and an estimated isothermal compressibility of $4.5 \times 10^{-5}$ bar$^{-1}$. Configurations were stored every 10 ps for subsequent analyses. Hydrogen bonds were identified using a geometric criterion. A hydrogen bond was logged if the donor-acceptor distance is within 0.25 nm and the donor-hydrogen-acceptor angle was larger than 135 degrees. The solvent-accessible surface area was determined by rolling a probe with diameter 0.14 nm over the surface of the protein, using slices of 0.005 nm width. An atom contact was assigned if the distance between two atoms were within 0.4 nm. The distributions of atom

contacts, hydrogen bonds and solvent-accessible surface area were estimated using a kernel density estimator with gaussian kernels. Distributions obtained from the first and second half of the simulations were compared to ensure convergence. Glycan densities were calculated using the program GROmaps (*Briones et al., 2019*).

## Recombinant expression of proteins

Soluble recombinant human ACE2 (rshACE2) was provided by Apeiron Biologicals (Vienna, Austria). Recombinant expression of all other proteins was performed by transient transfection of HEK293-6E cells, licensed from National Research Council (NRC) of Canada, as previously described (*Lobner et al., 2017*; *Durocher et al., 2002*). Cells were cultivated in FreeStyle F17 expression medium supplemented with 0.1% (v/v) Pluronic F-68 and 4 mM L-glutamine (all from Thermo Fisher Scientific, United States) in shaking flasks at 37 °C, 8% $CO_2$, 80% humidity and 130 rpm in a Climo-Shaker ISF1-XC (Adolf Kühner AG, Switzerland). pCAGGS vector constructs containing either the sequence of the SARS-CoV-2 RBD (residues R319-F541) or the complete luminal domain of Spike, modified in terms of removal of the polybasic furin cleavage site and introduction of two stabilizing point mutations (K986P and V987P), were kindly provided by Florian Krammer, Icahn School of Medicine at Mount Sinai (New York, United States) (*Amanat et al., 2020*; *Stadlbauer et al., 2020*). Plasmid constructs pcDNA3-sACE2(WT)-Fc(IgG1) and pcDNA3-sACE2-T92Q-Fc(IgG1) were obtained from Addgene (United States). The N322Q mutation was introduced into ACE2-wt-Fc and ACE2-T92Q-Fc using the QuikChange Lightning Site-Directed-Mutagenesis kit (Agilent Technologies, United States) according to the manufacturer's instructions and the respective parental vector as template. High-quality plasmid preparations for expression of ACE2-Fc variants were prepared using the PureYield Plasmid Midiprep System (Promega, United States). Transient transfection of the cells was performed at a cell density of approximately $1.7 \times 10^6$ cells mL$^{-1}$ culture volume using a total of 1 µg of plasmid DNA and 2 µg of linear 40 kDa polyethylenimine (Polysciences Inc, Germany) per mL culture volume. Forty-eight hr and 96 hr after transfection, cells were supplemented with 0.5% (w/v) tryptone N1 (Organotechnie, France) and 0.25% (w/v) D(+)-glucose (Carl Roth, Germany). Soluble proteins were harvested after 120–144 hr by centrifugation (10 000 g, 15 min, 4 °C).

## Purification of recombinantly expressed proteins

After filtration through 0.45 µm membrane filters (Merck Millipore, Germany), supernatants containing RBD or soluble Spike were concentrated and diafiltrated against 20 mM sodium phosphate buffer containing 500 mM NaCl and 20 mM imidazole (pH 7.4) using a Labscale TFF system equipped with a 5 kDa cut-off Pellicon XL device (Merck Millipore). The His-tagged proteins were captured using a 5 mL HisTrap FF crude column connected to an ÄKTA pure chromatography system (both from Cytiva, United States). Bound proteins were eluted by applying a linear gradient of 20–500 mM imidazole over 20 column volumes. ACE2-Fc variants were purified by affinity chromatography using a 5 mL HiTrap Protein A column (Cytiva) according to the manufacturer's instructions and 0.1 M glycine-HCl (pH 3.5) for elution. Eluate fractions were immediately neutralized using 2 M Tris (pH 12.0). Fractions containing the protein of interest were pooled, concentrated using Vivaspin 20 Ultrafiltration Units (Sartorius, Germany) and dialyzed against PBS (pH 7.4) at 4 °C overnight using SnakeSkin Dialysis Tubing (Thermo Fisher Scientific). The RBD was further purified by size exclusion chromatography (SEC) using a HiLoad 16/600 Superdex 200 pg column (Cytiva) eluted with PBS. All purified proteins were stored at –80 °C until further use.

## Enzymatic deglycosylation and desialylation of ACE2

For deglycosylation of ACE2-wt-Fc and rshACE2, proteins (2 mg mL$^{-1}$) were incubated with 180,000 U mL$^{-1}$ PNGase F (New England Biolabs, Unites States) in PBS (pH 7.4) for 24 hr at 37 °C. Desialylation of ACE2-wt-Fc was performed with 2500 U mL$^{-1}$ neuraminidase (New England Biolabs) in 50 mM sodium citrate (pH 5.0) under otherwise identical conditions. The deglycosylated or desialylated ACE2 variants were purified by preparative SEC using a HiLoad 16/600 Superdex 200 pg column eluted in PBS. The extent of enzymatic deglycosylation and desialylation was assessed by SDS-PAGE (*Figure 5— figure supplement 1*), SEC-MALS (*Figure 5—figure supplement 2*), and ESI-LC-MS/MS (*Figure 7— figure supplements 4 and 5*).

## Bio-layer interferometry (BLI) measurements

Interaction studies were performed on an Octet RED96e system using high precision streptavidin (SAX) biosensors (both from ForteBio, United States). Thus, all capture molecules (ACE2-wt-Fc, ACE2-T92Q-Fc, ACE2-N322Q-Fc, ACE2-T92Q-N322Q-Fc, deglyco-ACE2-wt-Fc, desialo-ACE2-wt-Fc, rshACE2, and deglyco-rshACE2) were biotinylated using the EZ-Link Sulfo-NHS-LC-Biotin kit (Thermo Fisher Scientific). Excess sulfo-NHS-LC-biotin was quenched by adding Tris-HCl buffer (800 mM, pH 7.4) to a final concentration of 3 mM. Biotinylated proteins were further purified using PD-10 desalting columns (Cytiva) according to the manufacturer's protocol. All assays were conducted in PBS supplemented with 0.05% (v/v) Tween 20% and 0.1% (w/v) BSA (PBST-BSA) at 25 °C with the plate shaking at 1000 rpm. The SAX biosensors were first equilibrated in PBST-BSA and then dipped into a 34 nM solution of the respective biotinylated capture molecule until a signal threshold of 0.8 nm was reached. Subsequently, the biosensors were dipped into PBST-BSA for 90 s to record a baseline, before they were submerged into different concentrations of RBD or the Spike protein to record association rates. For binding analysis of trimeric Spike, all biosensors were dipped into twofold serial dilutions of the protein (1.6–50 nM). To determine $K_D$ values, titration of RBD was performed at different concentrations to cover a broad concentration range around the respective $K_D$ value (*Hulme and Trevethick, 2010*). Biosensors loaded with ACE2 variants were submerged into twofold (6.25–200 nM) or threefold (0.8–200 nM) serial dilutions of RBD as appropriate for 600 s. For dissociation, the biosensors were dipped into PBST-BSA for 300 s (for analysis of Spike) or 100 s (for analysis of RBD). Each experiment included a baseline measurement using PBST-BSA (negative control) as well as a positive control (RBD). Of note, no unspecific binding of RBD or the Spike protein to SAX biosensors was observed. Data were evaluated under consideration of the limit of detection (LOD) and limit of quantification (LOQ) as reported elsewhere (*Armbruster and Pry, 2008*; *Carvalho et al., 2018*). Each experiment was performed three times. Analysis was performed using the Octet data analysis software version 11.1.1.39 (ForteBio) according to the manufacturer's guidelines.

## SDS-PAGE

SDS-PAGE was carried out using a 4–15% MINI-PROTEAN TGX Stain-Free Protein Gel, the Mini-PROTEAN Tetra Vertical Electrophoresis Cell (both from Bio-Rad Laboratories Inc, United States) and SDS-PAGE running buffer (20 mM Tris, 200 mM glycine, 0.1% (w/v) SDS). One µg of each purified protein was mixed with SDS sample buffer (62.5 mM Tris/HCl (pH 6.8), 2.5% (w/v) SDS, 10% (w/v) glycerol, 0.01% (w/v) bromophenol blue), heated to 70 °C for 10 min and loaded onto the gel. For reducing conditions purified samples were mixed with SDS-PAGE sample buffer containing 0.75 M β-mercaptoethanol and heated to 95 °C for 5 min prior to loading. The PageRuler Unstained Protein Ladder (Thermo Fisher Scientific) was used as a size marker. Protein bands were visualized with the Gel Doc XR + Imager (Bio-Rad Laboratories).

## Size-exclusion chromatography - multi-angle light scattering (SEC-MALS)

Size-exclusion chromatography combined with multi-angle light scattering was performed to determine the homogeneity and the native molecular mass of all proteins under study. Analyses were performed on an LC20 Prominence HPLC equipped with a refractive index detector RID-10A and the photodiode array detector SPD-M20A (all from Shimadzu, Japan). In-line MALS was analyzed either with a miniDAWN TREOS II MALS (for analysis of Spike) or a Heleos Dawn8+ plus QELS apparatus (Wyatt Technology, United States). Prior to analysis, all proteins were centrifuged (16,000 g, 10 min, 20 °C) and filtered (0.1 µm Ultrafree-MC filter, Merck Millipore). Proper performance of the MALS detectors was validated with bovine serum albumin. Purified Spike was analyzed by injection of a total of 50 µg onto a Superose 6 Increase 10/300 GL column (Cytiva) at a flow rate of 0.25 mL min$^{-1}$. The mobile-phase buffer used was PBS supplemented with 10% glycerol (pH 7.4). All other proteins were analyzed by using a Superdex 200 10/300 GL column (Cytiva) equilibrated with PBS plus 200 mM NaCl (pH 7.4). A total of 25 µg of each protein was injected and experiments were performed at a flow rate of 0.75 mL min$^{-1}$. Data were analyzed using the ASTRA six software (Wyatt Technology).

## Differential scanning calorimetry (DSC)

DSC experiments were performed using a MicroCal PEAQ-DSC Automated system (Malvern Panalytical, Germany), using 2.5 μM protein solutions in PBS (pH 7.4). The heating was performed from 20°C to 100°C at a rate of 1 °C/min. The protein solution was then cooled in situ and an identical thermal scan was run to obtain the baseline for subtraction from the first scan. All measurements were performed in triplicates. Fitting was done with Origin 7.0 for DSC software using the non-2-state transition model.

## Peptide mapping and glycopeptide analysis

All samples were analysed as in-solution proteolytic digests of the respective proteins by LC-ESI-MS/MS. For this, the pH of the samples was first adjusted to pH 7.8 by the addition of 1 M HEPES (pH 7.8) to a final concentration of 100 mM. The samples were then chemically reduced and S-alkylated, using a final concentration of 10 mM dithiothreitol for 30 min at 56 °C, and a final concentration of 20 mM iodoacetamide for 30 min at room temperature in the dark. To maximize sequence coverage, proteins were digested for 18 hr at 37 °C with chymotrypsin (Roche, Germany), followed by 3 hr at 37 °C using trypsin (Promega). All proteolytic digests were acidified to pH two by addition of 10% formic acid and directly analyzed by LC-ESI-MS/MS, using a capillary BioBasic C18 reversed-phase column (BioBasic-18, 150 × 0.32 mm, 5 μm, Thermo Fisher Scientific), installed in a Dionex Ultimate U3000 HPLC system (Thermo Fisher Scientific), developing a linear gradient from 95% eluent A (80 mM ammonium formate, pH 3.0, in HPLC-grade water) to 65% eluent B (80% acetonitrile in 80 mM ammonium formate, pH 3.0) over 50 min, followed by a linear gradient from 65% to 99% eluent B over 15 min, at a constant flow rate of 6 μL/min, coupled to a maXis 4 G Q-TOF instrument (Bruker Daltonics, Germany; equipped with the standard ESI source). For (glyco)peptide detection and identification, the mass spectrometer was operated in positive-ion DDA mode (i.e. switching to MS/MS mode for eluting peaks), recording MS scans in the m/z range from 150 to 2200 Th, with the six highest signals selected for MS/MS fragmentation. Instrument calibration was performed using a commercial ESI calibration mixture (Agilent Technologies). Site-specific profiling of protein glycosylation was performed using the dedicated Q-TOF data-analysis software packages Data Analyst (Bruker Daltonics) and Protein Scape (Bruker Daltonics), in conjunction with the MS/MS search engine MASCOT (Matrix Sciences Inc, United States) for automated peptide identification.

## ACE2 activity assays

Enzymatic activity of ACE2 was determined and quantified as described previously (*Vickers et al., 2002*), using 100 μM 7-methoxycoumarin-4-yl-acetyl-Ala-Pro-Lys-2,4-dinitrophenyl (Bachem, Switzerland) as substrate.

## SARS-CoV-2 neutralization assays

All work with infectious SARS-CoV-2 was performed under BSL-3 conditions. Vero E6 cells (ATCC, United States) were grown in Minimum Essential Medium (MEM) containing Earle's Salts, 1% penicillin/streptomycin stock solution and 2 mM L-glutamine (all from Thermo Fisher Scientific), supplemented with 5% fetal bovine serum (FBS), at 37 °C and 5% $CO_2$ and regularly tested for mycoplasma contaminations. A German 2019-nCoV isolate (Ref-SKU: 026 V-03883, Charité, Berlin, Germany) was propagated in Vero E6 cells. The TCID50 titer of virus stocks was determined by the Reed-Munch method (*Ramakrishnan, 2016*) and converted to plaque-forming units (pfu) using the conversion factor 0.7 (https://www.atcc.org/support/technical-support/faqs/converting-tcid-50-to-plaque-forming-units-pfu). Vero E6 cells were seeded in 48-well cell culture plates (3 × 10^4 cells per well) in MEM supplemented with 2% FBS overnight to reach approximately 80% confluence on the day of infection. ACE2 variants (final concentrations: 10–100 μg/mL) were preincubated with 60 pfu SARS-CoV-2 for 30 min at 37 °C under constant shaking (300 rpm). After preincubation, Vero E6 cells were infected for 1 h at 37 °C with samples containing either SARS-CoV-2 and ACE2 variants or solely SARS-CoV-2 (untreated controls) at a multiplicity of infection (MOI) of 0.002. Subsequently, cells were washed two times with MEM to remove unadsorbed virus. After incubation for 24 hr at 37 °C in MEM supplemented with 2% FBS, viral RNA was extracted from the culture supernatant using the QiaAmp Viral RNA Minikit (Qiagen, Germany), according to the manufacturer's protocol. SARS-CoV-2 replication was quantified via RT-qPCR using the QuantiTect Multiplex

RT-qPCR Kit (Qiagen) with a Rotor Gene Q cycler (Qiagen). The reactions were performed in a total volume of 25 µL at 50 °C for 30 min followed by 95 °C for 15 min and 45 cycles of 95 °C for 3 s and 55 °C for 30 s. Forward primer: 2019-nCoV_N1-F 5'-GACCCCAAAATCAGCGAAAT-3'; reverse primer: 2019-nCoV_N1-R 5'-TCTGGTTACTGCCAGTTGAATCTG-3'; probe: 2019-nCoV_N1-P 5'-FAM-ACCCCGCATTACGTTTGGTGGACC-BHQ1-3'. Statistical analyses were conducted using GraphPad Prism 8. Significance was determined by Kruskal-Wallis, comparing the mean rank of the ACE2-wt-Fc group with the mean rank of every other group (*, $p < 0.05$; **, $p < 0.01$; ***, $p < 0.001$).

SARS-CoV-2 neutralization assays were also performed independently in another laboratory using a different virus isolate. For these assays, Vero E6 cells were grown in Dulbecco's Modified Eagle's Medium (DMEM, Thermo Fisher Scientific) supplemented with 1% non-essential amino acid stock solution (Thermo Fisher Scientific), 10 mM HEPES (Thermo Fisher Scientific) and 10% FBS at 37 °C and 5% $CO_2$. SARS-CoV-2 isolated from a nasopharyngeal sample of a Swedish COVID-19 patient (GenBank accession number MT093571) was propagated in Vero E6 cells. Virus was titered using a plaque assay as previously described (*Becker et al., 2008*) with fixation of cells 72 hr post infection. Vero E6 cells were treated and infected as described previously (*Monteil et al., 2020*). Briefly, Vero E6 cells were seeded in 48-well plates ($5 \times 10^4$ cells per well) in DMEM containing 10% FBS. Twenty-four hr post-seeding, ACE2 variants (final concentrations: 50–200 µg/mL) were mixed with $10^6$ pfu SARS-CoV-2 (MOI: 20) in a final volume of 100 µl DMEM containing 5% FBS, incubated for 30 min at 37 °C and then added to the cells. Fifteen hr post-infection, cells were washed three times with PBS and then lysed using Trizol (Thermo Fisher Scientific) before analysis by RT-qPCR to quantify the content of SARS-CoV-2 RNA as described (*Monteil et al., 2020*).

## Acknowledgements

We thank Florian Krammer (Icahn School of Medicine at Mount Sinai, NY, United States) for providing the constructs used for production of recombinant Spike and RBD. pcDNA3-sACE2(WT)-Fc(IgG1) and pcDNA3-sACE2-T92Q-Fc(IgG1) were used with the kind permission of Erik Procko (University of Illinois, IL, United States). Transfection-grade pCAGGS plasmids were provided by Rainer Hahn and Gerald Striedner (University of Natural Resources and Life Sciences Vienna, Austria) in the frame-work of the BOKU COVID-19 Initiative. The authors thank Irene Schaffner and Jakob Wallner (BOKU Core Facility Biomolecular & Cellular Analysis) for assisting in BLI measurements, Gerhard Stadlmayr (University of Natural Resources and Life Sciences Vienna, Austria) for performing SEC-MALS analysis of Spike and ForteBio for supporting the BOKU COVID-19 Initiative with SAX biosensors. The high-performance computing center (HLRS) of the University of Stuttgart is gratefully acknowledged for providing computational resources. This project is supported by the PhD programme BioToP funded by the Austrian Science Fund (grant No. W1224-B09), Vienna Science and Technology Fund (WWTF; grant No. COV20-015), and funding from the Innovative Medicines Initiative 2 Joint Undertaking (JU; grant agreement No. 101005026). The JU receives support from the European Union's Horizon 2020 research and innovation programme and EFPIA. JWP is a recipient of a DOC Fellowship of the Austrian Academy of Sciences (ÖAW) at the Institute for Molecular Modeling and Simulation at the University of Natural Resources and Life Sciences, Vienna (Grant No. 24987). JMP and the research leading to these results has received funding from the T von Zastrow foundation, the FWF Wittgenstein award (grant No. Z271-B19), the Austrian Academy of Sciences, the Canada 150 Research Chairs Program in Functional Genetics (grant No. F18-0133), and the Canadian Institutes of Health Research COVID-19 grants F20-02343 and F20-02015.

## Additional information

### Competing interests

Janine Niederhöfer, Gerald Wirnsberger: employee of Apeiron Biologics. Apeiron holds a patent on the use of ACE2 for the treatment of lung, heart, or kidney injury and applied for a patent to treat COVID-19 with rshACE2. Josef M Penninger: declares a conflict of interest as a founder, supervisory board member, and shareholder of Apeiron Biologics. Apeiron holds a patent on the use of ACE2

for the treatment of lung, heart, or kidney injury and applied for a patent to treat COVID-19 with rshACE2. The other authors declare that no competing interests exist.

## Funding

| Funder | Grant reference number | Author |
|---|---|---|
| Austrian Science Fund | W1224-B09 | Daniel Maresch<br>Lukas Mach<br>Chris Oostenbrink<br>Nikolaus F Kienzl |
| Vienna Science and Technology Fund | COV20-015 | Tümay Capraz<br>Chris Oostenbrink |
| Innovative Medicines Initiative 2 Joint Undertaking | 101005026 | Vanessa Monteil<br>Ali Mirazimi<br>Josef M Penninger |
| DOC fellowship of the Academy of Sciences | 24987 | Jan W Perthold |
| T. von Zastrow foundation | | Josef M Penninger<br>Johannes Stadlmann |
| Austrian Science Fund | Z271-B19 | Josef M Penninger<br>Johannes Stadlmann |
| Canada Research Chairs | F18-0133 | Josef M Penninger |
| Canadian Institutes of Health Research | F20-02343 | Josef M Penninger |
| Canadian Institutes of Health Research | F20-02015 | Josef M Penninger |

The funders had no role in study design, data collection and interpretation, or the decision to submit the work for publication.

## Author contributions

Tümay Capraz, Elisabeth Laurent, Formal analysis, Investigation, Visualization, Writing – original draft; Nikolaus F Kienzl, Esther Föderl-Höbenreich, Formal analysis, Investigation, Visualization; Jan W Perthold, Investigation, Supervision, Writing – original draft; Clemens Grünwald-Gruber, Daniel Maresch, Vanessa Monteil, Janine Niederhöfer, Investigation; Gerald Wirnsberger, Investigation, Resources, Supervision; Ali Mirazimi, Kurt Zatloukal, Supervision; Lukas Mach, Formal analysis, Investigation, Supervision, Writing – original draft, Writing – review and editing; Josef M Penninger, Conceptualization, Writing – review and editing; Chris Oostenbrink, Conceptualization, Supervision, Writing – original draft, Writing – review and editing; Johannes Stadlmann, Conceptualization, Formal analysis, Investigation, Supervision, Writing – original draft, Writing – review and editing

## Author ORCIDs

Tümay Capraz ![ORCID] http://orcid.org/0000-0002-2547-067X
Nikolaus F Kienzl ![ORCID] http://orcid.org/0000-0001-8057-3930
Elisabeth Laurent ![ORCID] http://orcid.org/0000-0002-5234-5524
Jan W Perthold ![ORCID] http://orcid.org/0000-0002-8575-0278
Esther Föderl-Höbenreich ![ORCID] http://orcid.org/0000-0003-2066-1036
Clemens Grünwald-Gruber ![ORCID] http://orcid.org/0000-0002-6097-8348
Vanessa Monteil ![ORCID] http://orcid.org/0000-0002-2652-5695
Gerald Wirnsberger ![ORCID] http://orcid.org/0000-0001-7035-7038
Kurt Zatloukal ![ORCID] http://orcid.org/0000-0001-5299-7218
Lukas Mach ![ORCID] http://orcid.org/0000-0001-9013-5408
Josef M Penninger ![ORCID] http://orcid.org/0000-0002-8194-3777
Chris Oostenbrink ![ORCID] http://orcid.org/0000-0002-4232-2556
Johannes Stadlmann ![ORCID] http://orcid.org/0000-0001-5693-6690

## Decision letter and Author response

Decision letter https://doi.org/10.7554/eLife.73641.sa1
Author response https://doi.org/10.7554/eLife.73641.sa2

## Additional files

### Supplementary files

• Supplementary file 1. Glycosylation sites and N-glycan structures used in the models of SARS-CoV-2 Spike and human ACE2 are indicated. Blue squares indicate N-acetylglucosamine, green circles mannose, yellow circles galactose, purple squares sialic acid and green triangles fucose residues. Anomericity and positions of glycosidic linkages are indicated where necessary.

• Transparent reporting form

### Data availability

Molecular models and simulation trajectories are available through the BioExcel COVID-19 Molecular Structure and Therapeutics Hub (https://covid.bioexcel.eu/simulations/).

The following dataset was generated:

| Author(s) | Year | Dataset title | Dataset URL | Database and Identifier |
|---|---|---|---|---|
| Oostenbrink Lab | 2020 | Blocking SARS-CoV-2 Spike protein binding to human ACE2 receptor | https://covid.molssi.org//targets/#blocking-sars-cov-2-spike-protein-binding-to-human-ace2-receptor | covid.bioexcel.eu/simulations/, covid |

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
