## [Editor Report]

This paper takes on the important topic of therapeutics for SARS-CoV-2. This is an early-phase proof of concept study showing that a de-glycosylated form of the virus receptor, ACE2, could serve to reduce virus replication by acting as a decoy receptor that doesn't lead to infection.

---

## [Decision Letter]

**Decision letter after peer review:**

Thank you for submitting your article "Structure-guided glyco-engineering of ACE2 for improved potency as soluble SARS-CoV-2 decoy receptor" for consideration by *eLife*. Your article has been reviewed by 2 peer reviewers, and the evaluation has been overseen by a Reviewing Editor and Sara Sawyer as the Senior Editor. The reviewers have opted to remain anonymous.

Essential revisions:

It is important that you address all of the concerns before we can re-consider the paper at *eLife*. Close attention will be paid to your efforts to address all comments, but in particular the final comment by Reviewer 2: "Critically, only a 2-fold change in the apparent affinity between Spike and ACE2 are observed in Figure 6, and yet apparently orders magnitude difference in inhibitory activity in Figure 7. The implied causality between glycosylation and activity is not rigorously supported by the numbers."

*Reviewer #1 (Recommendations for the authors):*

Is there any toxicity in the cells due to deglycosylated ACE2 treatment?

Figure 7 – Confirmation of RNA levels with infectious virus measurements would strengthen the conclusions. In addition, a range of 2 to 3 experiments were used. It would be preferable if 3 experiments were used.

The SARS-CoV-2 stock used in these studies was propagated in Vero E6 cells, which means that the virus stock may contain a deletion in the spike cleavage site. The authors should confirm the results using Calu3 human lung cells.

*Reviewer #2 (Recommendations for the authors):*

There are two major issues with the molecular dynamics: (1) Why was this particular conformation of the Spike-ACE2 complex chosen for the analysis and not other possibilities (e.g., two "up")? This was not adequately explained. (2) The goal of the simulation was not explicitly stated, and this reviewer gets the impression that what was achieved was nothing more than what would be obtained from standard minimization procedures. These are important weakness of the paper, because all further analysis proceeds from the structural model after the simulation.

With regards to the post-simulation analysis, the authors do not go into sufficient detail on the data supporting the purported interactions between glycans and proteins. Proximity is obviously an insufficient argument, and even though data are presented in Figure 3 and Figure S5, these are not described in a way that facilitates assessment of the rigor and significance of the analysis. This also relates back to the molecular dynamics step: how are we to assess whether a 100 ns simulation sufficiently captures the range of conformations of these glycans?

On the functional data: while all of these experiments seem to support the conclusion that glycans are important, there are important discrepancies that remain unexplained. Critically, only a 2-fold change in the apparent affinity between Spike and ACE2 are observed in Figure 6, and yet apparently orders magnitude difference in inhibitory activity in Figure 7. The implied causality between glycosylation and activity is not rigorously supported by the numbers.

---

## [Author Response]

Essential revisions:It is important that you address all of the concerns before we can re-consider the paper at eLife. Close attention will be paid to your efforts to address all comments, but in particular the final comment by Reviewer 2: "Critically, only a 2-fold change in the apparent affinity between Spike and ACE2 are observed in Figure 6, and yet apparently orders magnitude difference in inhibitory activity in Figure 7. The implied causality between glycosylation and activity is not rigorously supported by the numbers."

All comments have been addressed in the revised manuscript or will be addressed by future experiments that will be linked to the manuscript. As pointed out by reviewer 2, deglycosylation of hACE2 improves SARS-CoV-2 neutralization more potently than anticipated from its effect on the affinity of hACE2 for the receptor-binding domain (RBD) of Spike as measured by BLI in vitro. We agree that this observation requires further explanation. As also stated in the revised manuscript, several of the following factors contribute to this apparent discrepancy between virus neutralization and RBD binding: (1) While the binding affinity is represented by an equilibrium constant (K_d_), which is defined as the hACE2 concentration at which 50% is in a bound state, the inhibitory activity is measured at a single concentration, at which one protein may still show hardly any inhibition (K_d_ is larger than the concentration in the assay) while another variant may show already strong inhibition (K_d_ is smaller than the concentration in the assay); (2) The BLI measurements were performed in a setting of a 1:1 interaction between hACE2 and monomeric RBD. in vitro, dimeric hACE2 binds to trimeric Spike, in which the RBDs are in a dynamic equilibrium of ‘down’ and ‘up’ states, leading to cooperative effects [Monod et al., J. Mol. Biol. (1965) 12, 88-118]; (3) A much higher degree of valency is relevant on the viral surface when hACE2 molecules need to block multiple Spike trimers, potentiating small differences in affinity. The inhibitory strength is not a function of the number of Spike proteins that are bound, but of the probability that a high enough fraction of Spike proteins is bound [Magnus, PLoS Comput. Biol. (2013) 9, e1002900]; (4) The SARS-CoV-2 neutralization assays shown in Figure 7 were performed at a low multiplicity of infection. This can lead to complete neutralization of all virus particles in the inoculum as shown in a previous study [Monteil et al., Cell (2020) 181, 905-913] and also evident from the individual data points of our experiments (Figure 7 —figure supplement 1). Hence, the data are presented as medians and not as arithmetic means, which potentially accentuates the numeric differences between individual samples.

Reviewer #1 (Recommendations for the authors):Is there any toxicity in the cells due to deglycosylated ACE2 treatment?

We thank the reviewer for this interesting question. Since native hACE2 did not show any cytotoxic effects even at a concentration as high as 200 µg/mL [Monteil et al., Cell (2020) 181, 905-913], we assume that treatment with 10-50 µg/mL deglycosylated hACE2 (as used in the experiments shown in Figure 7) will not lead to any notable cell death. In compliance with the current *eLife* revision policy, we now explicitly state in the revised manuscript that the lack of cellular toxicity by deglycosylated ACE2 still has to be experimentally verified and that these additional experiments will be carried out as soon as possible. The results will be linked to the original paper.

Figure 7 – Confirmation of RNA levels with infectious virus measurements would strengthen the conclusions. In addition, a range of 2 to 3 experiments were used. It would be preferable if 3 experiments were used.

We thank the reviewer for pointing out that we were not precise in our wording. With the exception of one single mutant (ACE2-T92Q-Fc), the data presented in Figure 7 are derived from 3 independent experiments. With ACE2-T92Q-Fc, only 2 experiments could be performed due to the limited availability of this protein. This is now clearly stated in the legend of Figure 7.

The SARS-CoV-2 stock used in these studies was propagated in Vero E6 cells, which means that the virus stock may contain a deletion in the spike cleavage site. The authors should confirm the results using Calu3 human lung cells.

We thank the reviewer for this suggestion. It is indeed known that loss of the furin cleavage site improves the fitness of SARS-CoV-2 in Vero E6 cells, whereas it reduces replication in Calu-3 cells [Johnson et al., Nature (2021) 591, 293-299]. Hence, we will verify the sequence of the furin cleavage site in our SARS-CoV-2 stock. Furthermore, we will repeat our Vero E6 results with Calu-3 cells and the SARS-CoV-2 Δ variant, the increased infectivity of which has been linked to its enhanced furin susceptibility [Planas et al., Nature (2021), 596, 276-280]. This will be done in additional experiments of highest priority. The results will be linked to the original paper as soon as possible.

Reviewer #2 (Recommendations for the authors):There are two major issues with the molecular dynamics: (1) Why was this particular conformation of the Spike-ACE2 complex chosen for the analysis and not other possibilities (e.g., two "up")? This was not adequately explained.

In this work, we have focused on a single interface between the Spike trimer and the hACE2 dimer. The reviewer is correct that there are also configurations of the Spike trimer, with two (or three) receptor binding domains (RBDs) in the “up” conformation. Using this model would be of interest if it is expected that both RDBs would be able to simultaneously interact with both monomers of the hACE2 dimer. As this is sterically very unlikely, we chose to use the Spike with a single RBD in the “up” conformation. Cryo-EM structures with two RBDs in the up conformation were seen to bind to two separate molecules of hACE2, rather than to a single ACE2 dimer [Benton et al., Nature (2020) 588, 327–330]. Using such complexes would unnecessarily complicate our analysis of the effect of glycans on the Spike / hACE2 interactions. We have clarified this in the revised manuscript.

(2) The goal of the simulation was not explicitly stated, and this reviewer gets the impression that what was achieved was nothing more than what would be obtained from standard minimization procedures. These are important weakness of the paper, because all further analysis proceeds from the structural model after the simulation.

We agree with the reviewer that the overall protein structures did not change much beyond what could be obtained with energy minimizations from the experimentally determined structures. However, for the glycans, this is not the case. The glycan conformations were not observed experimentally and were initially modelled in a low-energy conformation. As can be seen from Figure 4, the molecular dynamics simulations subsequently allow for substantial sampling of the conformational space of the glycans. A single configuration, as obtained from an energy minimization, would not have yielded such diverse distributions of interactions. Furthermore, we would not have been able to make the comparison in Figure 4, where, in the absence of Spike, the glycans are shown to cover the Spike binding site. We emphasize this point in the revised manuscript.

With regards to the post-simulation analysis, the authors do not go into sufficient detail on the data supporting the purported interactions between glycans and proteins. Proximity is obviously an insufficient argument, and even though data are presented in Figure 3 and Figure S5, these are not described in a way that facilitates assessment of the rigor and significance of the analysis. This also relates back to the molecular dynamics step: how are we to assess whether a 100 ns simulation sufficiently captures the range of conformations of these glycans?

While we agree that proximity is a poor representation of the strength of the interactions, we think it offers a simple characterization of the interactions between the glycans and the Spike protein. Our analysis in Figure 4 suggests that the glycans cover the interface area of hACE2 in the absence of Spike, while they contribute to about 50% of the overall interface area upon binding of Spike. We have extended our analysis of the data in Figure 3 of the revised manuscript. To address the question if the 100 ns of simulations are sufficient, we have split our analyses of the two independent 100-ns simulations into four parts of 50 ns each, and report the data from Figure 3 for the individual time-segments in a new Figure 3 —figure supplement 1. As discussed in the original manuscript, the glycan at N546 interacted with Spike for a significant amount of time only in the first of the two independent simulations. In contrast, the glycan at N90 seems to be showing more interactions with Spike in the second simulation than in the first simulation. For the rest, the trends seem to be quite similar for the individual simulations, as well as when analyzing only their first or second halves. We comment on this additional analysis in the revised manuscript.

On the functional data: while all of these experiments seem to support the conclusion that glycans are important, there are important discrepancies that remain unexplained. Critically, only a 2-fold change in the apparent affinity between Spike and ACE2 are observed in Figure 6, and yet apparently orders magnitude difference in inhibitory activity in Figure 7. The implied causality between glycosylation and activity is not rigorously supported by the numbers.

We agree that deglycosylation of hACE2 improves SARS-CoV-2 neutralization more potently than anticipated from its effect on the affinity of hACE2 for RBD as measured by BLI in vitro. Several of the following factors contribute to this apparent discrepancy between virus neutralization and RBD binding: (1) While the binding affinity is represented by an equilibrium constant (K_d_), which is defined as the hACE2 concentration at which 50% is in a bound state, the inhibitory activity is measured at a single concentration, at which one protein may still show hardly any inhibition (K_d_ is larger than the concentration in the assay) while another variant may show already strong inhibition (K_d_ is smaller than the concentration in the assay); (2) The BLI measurements were performed in a setting of a 1:1 interaction between hACE2 and monomeric RBD. in vitro, dimeric hACE2 binds to trimeric Spike, in which the RBDs are in a dynamic equilibrium of ‘down’ and ‘up’ states, leading to cooperative effects [Monod et al., J. Mol. Biol. (1965) 12, 88-118]; (3) A much higher degree of valency is relevant on the viral surface when hACE2 molecules need to block multiple Spike trimers, potentiating small differences in affinity. The inhibitory strength is not a function of the number of Spike proteins that are bound, but of the probability that a high enough fraction of Spike proteins is bound [Magnus, PLoS Comput. Biol. (2013) 9, e1002900]; (4) The SARS-CoV-2 neutralization assays shown in Figure 7 were performed at a low multiplicity of infection. This can lead to complete neutralization of all virus particles in the inoculum as shown in a previous study [Monteil et al., Cell (2020) 181, 905-913] and also evident from the individual data points of our experiments (Figure 7 —figure supplement 1). Hence, the data are presented as medians and not as arithmetic means, which potentially accentuates the numeric differences between individual samples.